# HETEROGENEOUS SWARMS: Jointly Optimizing Model Roles and Weights for Multi-LLM Systems

**Shangbin Feng**[1]* **Zifeng Wang**[2] **Palash Goyal**[2] **Yike Wang**[1] **Weijia Shi**[1]
**Huang Xia**[3] **Hamid Palangi**[2] **Luke Zettlemoyer**[1] **Yulia Tsvetkov**[1] **Chen-Yu Lee**[2] **Tomas Pfister**[2]
[1]University of Washington  [2]Google Cloud AI Research  [3]Google
shangbin@cs.washington.edu {zifengw,chenyulee}@google.com

## Abstract

We propose HETEROGENEOUS SWARMS, an algorithm to design multi-LLM systems by jointly optimizing model roles and weights. We represent multi-LLM systems as directed acyclic graphs (DAGs) of LLMs with topological message passing for collaborative generation. Given a pool of LLM experts and a utility function, HETEROGENEOUS SWARMS employs two iterative steps: *role-step* and *weight-step*. For *role-step*, we interpret model roles as learning a DAG that specifies the flow of inputs and outputs between LLMs. Starting from a swarm of random continuous adjacency matrices, we decode them into discrete DAGs, call the LLMs in topological order, evaluate on the utility function (e.g. accuracy on a task), and optimize the adjacency matrices with particle swarm optimization based on the utility score. For *weight-step*, we assess the contribution of individual LLMs in the multi-LLM systems and optimize model weights with swarm intelligence. We propose *JFK-score* to quantify the individual contribution of each LLM in the best-found DAG of the role-step, then optimize model weights with particle swarm optimization based on the JFK-score. Experiments demonstrate that HETEROGENEOUS SWARMS outperforms 17 role- and/or weight-based baselines by 18.5% on average across 12 tasks. Further analysis reveals that HETEROGENEOUS SWARMS discovers multi-LLM systems with heterogeneous model roles and substantial collaborative gains, and benefits from the diversity of language models.[2]

## 1 Introduction

Advancing beyond training a single general-purpose LLM [6, 87], recent research recognizes the importance of multi-LLM collaboration and advances *multi-LLM systems*, where diverse models serve in a collaborative system to complement each other and expand model capabilities [58, 79]. Models often have different *roles* in multi-LLM collaboration, governing the subtask and functionality of individual LLMs; adapting the model *weights* of these LLMs are also identified as important for models to complement each other. Existing methods to develop multi-LLM systems are often *fixed-weight* and/or *fixed-role* and could not flexibly adapt to diverse tasks and contexts.

*Fixed-weight* systems employ static and often black-box LLMs and contextualize their roles through textual inter-

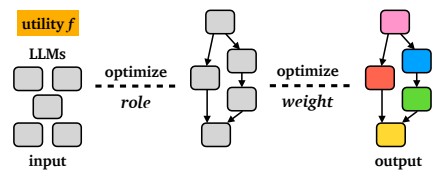

Figure 1: Our objective: given a pool of LLMs and a task utility function $f$, discover a multi-LLM system with graph-based model roles and adapted model weights tailored to $f$.

---

*Work done as a student researcher at Google Cloud AI Research.

[2]Resources available at https://github.com/BunsenFeng/heterogeneous_swarm.

39th Conference on Neural Information Processing Systems (NeurIPS 2025).

**Algorithm 1:** Particle Swarm Optimization step (PSO)

---

**Input:** vectors $\{\mathbf{x}_i\}_{i=1}^n$ and the utility values $\{f(\mathbf{x}_i)\}_{i=1}^n$ by utility function $f$

**Hyperparameters**: step length $\lambda$, inertia $\phi_v$, cognitive coeff. $\phi_p$, social coeff. $\phi_g$, repel coeff. $\phi_w$

**State variables**: each vector $\mathbf{x}_i$ has velocity $\mathbf{v}_i$ and personal best $\mathbf{p}_i$, global best and worst $\mathbf{g}$ and $\mathbf{g}_w$

**for** $i = 1$ **to** $n$ parallel **do**

    walk randomness $r_v, r_p, r_g, r_w \sim \text{Uniform}(0,1)$

    $\mathbf{v}_i \leftarrow \frac{1}{\mathcal{C}} \left[ r_v \phi_v \mathbf{v}_i + r_p \phi_p (\mathbf{p}_i - \mathbf{x}_i) + r_g \phi_g (\mathbf{g} - \mathbf{x}_i) - r_w \phi_w (\mathbf{g}_w - \mathbf{x}_i) \right]$, where normalization

    term $\mathcal{C} = r_v \phi_v + r_p \phi_p + r_g \phi_g + r_w \phi_w$

    $\mathbf{x}_i \leftarrow \mathbf{x}_i + \lambda \mathbf{v}_i$

**end**

Update personl/global information $\mathbf{p}_i$, $\mathbf{g}$, and $\mathbf{g}_w$

**return** $\{\mathbf{x}_i\}_{i=1}^n$, $\mathbf{x}_{best} = \arg\max_{\mathbf{x}} f(\mathbf{x})$

---

action [21, 120]. Despite the diversity of tasks and inputs, these static models are repeated across model roles and contexts, becoming the bottleneck of flexible adaptation. *Fixed-role* systems usually orchestrate LLMs in a fixed workflow and seed LLMs with different hand-crafted prompts and enable their interaction of message passing based on these prompt-induced roles [80, 25]. However, new tasks and domains would require substantial prompt engineering, which heavily depends on prior knowledge of the given task. Hence the static roles become the bottleneck in automating and scaling multi-LLM systems to unseen tasks and contexts [49, 88, 50]. As such, a flexible approach that jointly optimizes the weights and roles of multi-LLM systems is crucial for adapting diverse LLM experts for wide-ranging purposes.

We propose HETEROGENEOUS SWARMS to search for adapted roles and weights guided by utility function $f$ (*e.g.* performance on a task) via swarm intelligence. Inspired by the success of LLMs and particle swarm optimization (PSO) [48, 26], an algorithm to optimize continuous tensors via collective search, we employ two *interleaved* steps: *role-step* and *weight-step*.

For role-step, we use the heterogeneous expertise of models for varying roles: we interpret roles as input-output relations in a multi-LLM system. Role optimization is a graph learning problem, where natural language messages are passed between LLMs organized in a directed acyclic graph (DAG). While previous methods considered heuristics-based and hand-crafted structures such as a star or chain [52], finding the optimal structure of multi-LLM systems requires task-specific adaptation [37]. We use particle swarm optimization to learn LLM DAGs: Given a set of $n$ LLM experts, we randomly initialize a swarm of continuous adjacency matrices of size $n \times n$, indicating the likelihood of having directed edge $(i, j)$. We propose G-DECODE to *decode* these continuous adjacency matrices into discrete DAGs of models. We call LLMs in the topological order of the DAG and evaluate their performance: the swarm of adjacency matrices are then optimized based on the performance scores via PSO, where matrices collectively "move" in the matrix search space to adapt to $f$.

For weight-step, we adapt models to the task and roles represented by the multi-LLM network. We propose *JFK-score* to quantify *what individual models can do for the multi-LLM system*: assigning the pool of LLMs to positions in the DAG multiple times, run inference, and evaluate different assignments. The *JFK-score* of one model is the aggregated performance across assignments weighted by their frequency (i.e. number of times it appears in a multi-LLM system). The swarm of LLMs is then optimized based on the JFK-scores via PSO, where models update their weights to adapt to $f$.

Role-step and weight-step are iteratively employed to jointly optimize roles and weights, until the utility $f$ does not improve or a maximum iteration limit is met. In the end, we obtain a multi-LLM network where the graph structure and model weights are both optimized for utility $f$. (Figure 1)

HETEROGENEOUS SWARMS outperforms 15 role/weight-based approaches by 18.5% on average across 12 tasks spanning knowledge, reasoning, and agent contexts. HETEROGENEOUS SWARMS also enables inference-time scaling [82, 5, 100] of smaller language models through topological collaborative generation, with a 27.1% improvement on average when scaling the collaboration from two to ten LLMs. (Figure 6) Further analysis reveals that roles and weights could have varying levels of importance for different tasks, HETEROGENEOUS SWARMS benefits from the diversity of LLM

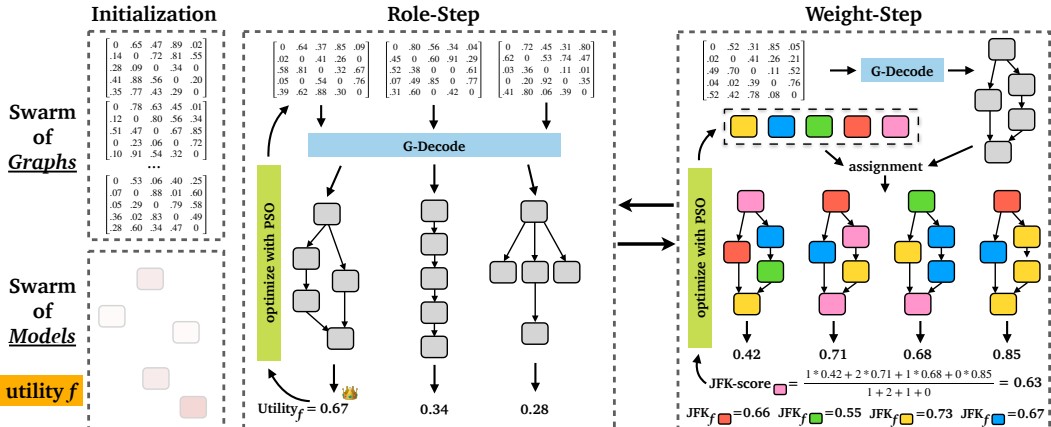

Figure 2: Overview of HETEROGENEOUS SWARMS: starting with a swarm of graphs represented by continuous adjacency matrices and a swarm of LLMs, HETEROGENEOUS SWARMS rotates between role-step and weight-step. In the role-step, we decode continuous adjacency adjacencies into discrete graphs, call the LLMs in topological order to fulfill a task, evaluate on the utility function, and optimize the adjacency matrices with particle swarm optimization. In the weight-step, models are randomly assigned to positions in the best-found network, evaluated by their individual contribution through the JFK-Score, and then optimized with particle swarm optimization. PSO denotes particle swarm optimization (Sec 2), G-Decode denotes Algorithm 2, and $f$ denotes the utility function.

experts, the optimization could speed up with increased sparsity and "dropout", and we discover multi-LLM systems with heterogeneous roles and collaborative gains.

## 2 Preliminary

**Multi-LLM Systems** Multi-LLM collaboration could take many forms, featuring information exchange at the API [36], text [21], logit [58], and weight levels [102]. In this work, we define multi-LLM systems as directed acyclic graphs (DAG) of LLMs and a multi-LLM system is represented by two variables: *a set of LLMs* $\mathcal{X} = [\mathbf{x}_1, \cdots, \mathbf{x}_n]$ and *adjacency matrix* $\mathcal{A} = \{a_{ij}\}_{n \times n}$, where $a_{ij} \in \{0, 1\}$ and $a_{ij} = 1$ indicates that the output text of model $\mathbf{x}_i$ becomes part of the input of model $\mathbf{x}_j$ with $n$ models in total. $\mathcal{A} \mid \mathcal{X}$ then denotes a multi-LLM system, where $\mathcal{A}$ defines the structure of the DAG and each position in the graph is instantiated by an LLM in $\mathcal{X}$. Given an input, models in $\mathcal{X}$ are called in the topological order of $\mathcal{A}$ to interact and collaborate with each other. As such, optimizing *roles* and input-output relations becomes optimizing the graph structure $\mathcal{A}$, and optimizing *weights* becomes optimizing the weights of models in $\mathcal{X}$.

**Swarm Intelligence** Particle swarm optimization (PSO) is an optimization algorithm based on the collective behavior of individual systems for non-differentiable contexts [48]. Guided by its initial success in LLMs [26], we adapt PSO to optimize swarms of $\mathcal{A}$ and $\mathbf{x}$.

The input of PSO is a set of continuous vectors $\{\mathbf{x}_i\}_{i=1}^n$ and a utility function $f$ giving vectors a scalar score $f(\mathbf{x})$. Each vector $\mathbf{x}_i$ has two attributes: velocity $\mathbf{v}_i$ of the same dimension, initialized as $\mathbf{0}$; personal best $\mathbf{p}_i$, the best-found location of $\mathbf{x}_i$ based on utility function $f$ in its search history. The collective swarm has two other attributes: global best/worst $\mathbf{g}$ and $\mathbf{g}_w$, indicating the best/worst-found checkpoint in all of the previous $\{\mathbf{x}_i\}_{i=1}^n$. These attributes would provide signals for vector $\mathbf{x}_i$ to move in the search space guided by personal/global information.

One PSO step changes model velocity $\mathbf{v}_i$ and takes a step towards the adjusted velocity direction:

$$\mathbf{v}_i \leftarrow \frac{1}{\mathcal{C}} \big[ r_v \phi_v \mathbf{v}_i + r_p \phi_p (\mathbf{p}_i - \mathbf{x}_i) + \\ r_g \phi_g (\mathbf{g} - \mathbf{x}_i) - r_w \phi_w (\mathbf{g}_w - \mathbf{x}_i) \big]$$

where $\mathcal{C} = r_v \phi_v + r_p \phi_p + r_g \phi_g + r_w \phi_w$ is a normalization term. The adjusted velocity is the weighted average of four terms: $\mathbf{v}_i$ makes the model keep part of its current velocity (i.e. inertia); $(\mathbf{p}_i - \mathbf{x}_i)$ draws the model towards its personal best; $(\mathbf{g} - \mathbf{x}_i)$ draws towards the global best; $-(\mathbf{g}_w - \mathbf{x}_i)$, repel from the global worst. $\phi_v, \phi_p, \phi_g, \phi_w$ are hyperparameters and $r_v, r_p, r_g, r_w \sim \mathcal{U}(0, 1)$ are randomness factors, governing how much $\mathbf{x}_i$ is impacted by personal/global information. Vectors

then take a step towards the new velocity: $\mathbf{x}_i \leftarrow \mathbf{x}_i + \lambda \mathbf{v}_i$, where $\lambda$ is the step length hyperparameter. New vectors are then evaluated on $f$ to update personal/global best and worst. We formulate a PSO step in algorithm 1 and will use $\{\mathbf{x}_i\}_{i=1}^n, \mathbf{x}_{best} = \text{PSO}(\{\mathbf{x}_i\}_{i=1}^n, \{f(\mathbf{x}_i)\}_{i=1}^n)$ to denote a PSO step to optimize $\{\mathbf{x}_i\}_{i=1}^n$ based on their scores $\{f(\mathbf{x}_i)\}_{i=1}^n$.

## 3 Methodology

We propose HETEROGENEOUS SWARMS, jointly optimizing model roles and weights for multi-LLM systems. We employ PSO for the optimization, since it uniquely enables us to leverage *multiple* diverse collaboration patterns and specialized model checkpoints, composing their expertise and strengths in multi-LLM systens. Starting from a pool of LLM experts as input, HETEROGENEOUS SWARMS discovers a directed acyclic graph of models representing heterogeneous roles with adapted model weights, iteratively employing *role-step* and *weight-step* (Figure 2).

### 3.1 Role-step

Unlike existing work that often manually specifies model roles through prompt engineering [25, 116], we propose to learn a directed acyclic graph of *input-output relationships*. To this end, HETEROGE-NEOUS SWARMS operationalizes the optimization of model roles as a graph optimization problem: discovering and optimizing directed acyclic graphs (DAGs) of LLMs to call them in the topological order for collaborative generation. We envision swarm intelligence as a viable and flexible tool by letting multiple potential DAGs explore the search space of graphs to adapt to a given task and utility function.

To make *discrete* graphs compatible with particle swarm optimization (Sec 2) that optimizes *continuous* tensors, we first represent graphs by continuous adjacency matrices $\mathbf{A} \in \mathbb{R}^{n \times n}$, where $a_{ij}$ denotes the likelihood of a directed edge from model $\mathbf{x}_i$ to model $\mathbf{x}_j$. We propose G-decode, an algorithm to decode discrete DAGs out of continuous $\mathbf{A}$s. We first select an end node $k$ based on inverse out degrees $k = \text{top-p}(\{1/\sum_{j=1}^n a_{ij}\}_{i=1}^n)$, where top-p denotes top-p sampling [35]. We then iteratively select a remaining node based on out degrees $u = \text{top-p}(\{\sum_{j=1}^n a_{ij}\})$ and add an edge between $u$ and any existing node $v$ with probability $\frac{\exp(a_{uv})}{\sum_{i \in \mathcal{E}} \exp(a_{ui})}$, until all nodes are considered. Graphs decoded with G-decode are directed since we only add directed edges and they are acyclic since new nodes only connect to existing nodes in the graph. The resulted DAGs define an input-output mapping: given an input, we call LLMs in the topological order of G-decode($\mathbf{A}$), and the output of the end node becomes the output of the multi-LLM system (Algorithm 4).

We evaluate the utility of the continuous adjacency matrix by decoding it into a DAG, call the models in topological order, and evaluate its performance on the utility function $f$. Concretely, we initialize a swarm of $N$ continuous adjacency matrices $\{\mathbf{A}^i\}_{i=1}^N \sim \mathcal{U}_{n \times n}(0, 1)$. For one role-step, we decode and evaluate their utility $f(\text{G-decode}(\mathbf{A}^i))$ and optimize the continuous $\mathbf{A}$s through swarm intelligence:

$$\{\mathbf{A}^i\}_{i=1}^N, \mathbf{A}_{best} \leftarrow \text{PSO}(\{\mathbf{A}^i\}_{i=1}^N, \{f(\text{G-decode}(\mathbf{A}^i))\}_{i=1}^N)$$

The resulting $\{\mathbf{A}^i\}_{i=1}^N$ after a single PSO step will represent graphs that slightly better adapt to the task $f$.

### 3.2 Weight-step

While existing work often uses static models across tasks and contexts [120, 21], we posit that model weights could be optimized to adapt to the task as well as model roles in the DAG. However, it is challenging to quantify the utility of a single LLM in a multi-LLM system, so that model weight optimization is compatible with swarm intelligence. We propose *JFK-score*[2], an algorithm to evaluate individual contribution in multi-LLM systems.

Concretely, given the best-found DAG in the role-step $\mathbf{A}_{best}$ and the pool of LLM experts $\{\mathbf{x}_i\}_{i=1}^n$, we randomly select an LLM for each position in $\mathbf{A}_{best}$ to obtain an assignment $\mathcal{X}$, where $\mathbf{A}_{best} \mid \mathcal{X}$

---

[2] *"Ask not what the multi-LLM system can do for you, ask what you can do for the multi-LLM system."* —— Authors, 2025

**Algorithm 2:** JFK-score

**Input:** adjacency matrix $\mathbf{A}_{best}$ and models $\{\mathbf{x}_i\}_{i=1}^n$
**for** $i = 1$ **to** $M$ **do**
> sample assignment $\mathcal{X}^i$ by randomly select models in $\{\mathbf{x}_i\}_{i=1}^n$ to fill each position in $\mathbf{A}_{best}$
> utility $f(\mathcal{X}^i) = f(\text{G-decode}(\mathbf{A}_{best} \mid \mathcal{X}^i))$

**end**
**for** $i = 1$ **to** $n$ **do**
> $\text{JFK-score}(\mathbf{x}_i) = \frac{\sum_{j=1}^M \text{cnt}_{i,j} \times f(\mathcal{X}^i)}{\sum_{j=1}^M \text{cnt}_{i,j}}$, $\text{cnt}_{i,j}$ denotes the frequency of $\mathbf{x}_i$ in assignment $\mathcal{X}^j$

**end**
**return** $\{\text{JFK-score}(\mathbf{x}_i)\}_{i=1}^n$

---

**Algorithm 3:** Heterogeneous Swarms

**Input:** language models $\{\mathbf{x}_i\}_{i=1}^n$, utility function $f$
sample $\{\mathbf{A}^i\}_{i=1}^N \sim \mathcal{U}_{n \times n}(0, 1)$
**while** $f$ *is improving* **do**
> // role-step
> $\{\mathbf{A}^i\}_{i=1}^N, \mathbf{A}_{best} \leftarrow \text{PSO}(\{\mathbf{A}^i\}_{i=1}^N, \{f(\text{G-decode}(\mathbf{A}^i))\}_{i=1}^N)$
> // weight-step
> $\{\mathbf{x}_i\}_{i=1}^n, \mathbf{x}_{best} \leftarrow \text{PSO}(\{\mathbf{x}_i\}_{i=1}^n, \{\text{JFK-score}(\mathbf{x}_i)\}_{i=1}^n)$

**end**
**return** $\text{G-decode}(\mathbf{A}_{best} \mid \{\mathbf{x}_i\}_{i=1}^n)$

---

represents an instantiated multi-LLM system. We repeat assignment $M$ times to obtain $\{\mathcal{X}^i\}_{i=1}^M$ and evaluate their utility $f(\mathcal{X}^i) = f(\text{G-decode}(\mathbf{A}^i \mid \mathcal{X}^i))$. The individual contribution of model $\mathbf{x}_i$ should then be the aggregate of utility weighted by model $i$'s frequency: we denote the frequency of $\mathbf{x}_i$ in $\mathcal{X}^j$ as $\text{cnt}_{i,j} = \sum_{k=1}^n \mathbb{1}(\mathcal{X}_k^j = \mathbf{x}_i)$ and the individual contribution score should be:

$$\text{JFK-score}(\mathbf{x}_i) = \frac{\sum_{j=1}^n \text{cnt}_{i,j} \times f(\mathcal{X}^i)}{\sum_{j=1}^n \text{cnt}_{i,j}}$$

In this way, $\text{JFK-score}(\mathbf{x}_i)$ quantifies the individual contribution of model $\mathbf{x}_i$ across multiple roles and collaboration partners. (Algorithm 2) We optimize model weights guided by their individual contribution with swarm intelligence:

$$\{\mathbf{x}_i\}_{i=1}^n, \mathbf{x}_{best} \leftarrow \text{PSO}(\{\mathbf{x}_i\}_{i=1}^n, \{\text{JFK-score}(\mathbf{x}_i)\}_{i=1}^n)$$

HETEROGENEOUS SWARMS alternates between role-step and weight-step, iteratively optimizing model roles and weights in the multi-LLM system to adapt to the task $f$. We present the overall procedure in algorithm 3.

## 4 Experiment Settings

**Models and Implementation** We implement a prototype of HETEROGENEOUS SWARMS with GEMMA-7B (*google/gemma-7b-it*) [29] in the main paper and also employ other LLMs such as MISTRAL-7B in Table 5. We employ the pool of 10 LLM experts in Feng et al. [26] for fair comparison, fine-tuned from GEMMA-7B using 10 domains in Tulu-v2 [44] spanning reasoning, code, general instruction following, and more: We employ $p = 0.8$ for top-p sampling, $N = 10$, $M = 10$, search patience 6, max iteration 20, while running grid search over other hyperparameters and report performance of the best-found multi-LLM systems.

**Baselines** We compare with 17 baselines across 5 categories that focus on optimizing model roles and/or weights.

| | Knowledge | | | Reasoning | | | Agent | | | Miscellaneous | | |
|---|---|---|---|---|---|---|---|---|---|---|---|---|
| | MMLU-pro | K-Cross | COM2 | GSM8k | NLGraph | Normad | GAIA-text | AB-kg | AB-ltp | Qasper | AbstainQA | WoW |
| BEST SINGLE | 0.231 | 0.346 | 0.488 | 0.237 | 0.535 | 0.545 | 0.107 | 0.383 | 0.120 | 0.174 | 0.065 | 0.415 |
| PRED. MERGE | 0.173 | 0.309 | 0.391 | 0.074 | 0.502 | 0.481 | 0.036 | 0.225 | / | / | / | 0.471 |
| DATA MERGE | 0.176 | 0.370 | 0.377 | 0.143 | 0.423 | 0.415 | 0.071 | 0.242 | 0.112 | 0.147 | -0.025 | 0.461 |
| UNIFORM SOUP | 0.206 | 0.295 | 0.519 | 0.352 | 0.500 | 0.430 | 0.036 | 0.392 | 0.105 | 0.166 | 0.003 | 0.455 |
| DARE-TIES | 0.230 | 0.372 | 0.476 | 0.307 | 0.544 | 0.427 | 0.071 | 0.300 | 0.108 | 0.137 | 0.140 | 0.515 |
| GREEDY SOUP | 0.219 | 0.355 | 0.539 | 0.330 | 0.530 | 0.543 | 0.071 | 0.333 | 0.114 | 0.184 | 0.014 | 0.565 |
| PACK OF LLMs | 0.235 | 0.352 | 0.512 | 0.327 | 0.532 | 0.543 | 0.143 | 0.392 | 0.106 | 0.157 | 0.095 | 0.545 |
| LORAHUB | 0.231 | 0.291 | 0.502 | 0.354 | 0.568 | 0.548 | 0.071 | 0.375 | 0.106 | 0.169 | 0.064 | 0.530 |
| MODEL SWARMS | 0.254 | 0.428 | 0.505 | 0.459 | **0.672** | 0.554 | 0.107 | 0.358 | 0.135 | 0.225 | 0.175 | 0.540 |
| CHAIN | 0.216 | 0.310 | 0.495 | 0.295 | 0.462 | 0.489 | 0.143 | 0.325 | 0.148 | 0.218 | 0.014 | 0.493 |
| STAR | 0.250 | 0.342 | 0.508 | 0.333 | 0.545 | 0.518 | 0.036 | 0.283 | 0.130 | 0.216 | 0.125 | 0.499 |
| GPT-SWARM | 0.216 | 0.320 | 0.460 | 0.334 | 0.611 | 0.510 | 0.143 | 0.333 | 0.134 | 0.216 | 0.023 | 0.492 |
| META-AGENT | 0.212 | 0.276 | 0.477 | 0.433 | 0.515 | 0.369 | 0.071 | 0.325 | 0.112 | 0.167 | 0.016 | 0.472 |
| AGENT-PRUNE | 0.214 | 0.321 | 0.497 | 0.180 | 0.460 | 0.467 | 0.107 | 0.333 | 0.122 | 0.211 | -0.005 | 0.470 |
| GNNs | 0.201 | 0.339 | 0.479 | 0.364 | 0.593 | 0.530 | 0.071 | 0.308 | 0.148 | 0.203 | 0.076 | 0.503 |
| AGENTVERSE | 0.239 | 0.309 | 0.501 | 0.403 | 0.633 | 0.513 | 0.107 | 0.367 | 0.136 | 0.195 | 0.103 | 0.489 |
| MACNET | 0.252 | 0.323 | 0.517 | 0.409 | 0.617 | 0.537 | 0.143 | 0.383 | 0.138 | 0.207 | 0.127 | 0.512 |
| H-SWARMS | **0.312** | **0.450** | **0.579** | **0.481** | 0.660 | **0.588** | **0.250** | **0.425** | **0.215** | **0.266** | **0.220** | **0.590** |

Table 1: Performance on the 12 datasets, best in **bold** and second-best in underline. HETEROGENEOUS SWARMS outperforms TRIVIAL, STATIC WEIGHT, DYNAMIC WEIGHT, STATIC ROLE, and DYNAMIC ROLE approaches by 18.5% on average across tasks.

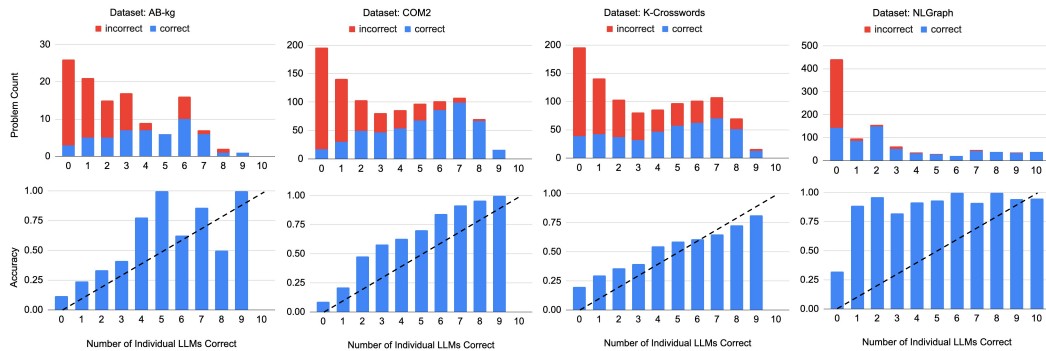

Figure 3: Evaluating collaborative gains: we create problem buckets by how many out of the 10 individual LLMs could solve it correctly. Top row: the problem count as well as whether the multi-LLM correctly solves the problems in each bucket. Bottom row: accuracy of the multi-LLM system in each bucket and expected accuracy denoted by the dotted line. HETEROGENEOUS SWARMS achieves collaborative gains (C-Gain) of 0.143, 0.184, 0.101, and 0.426 on the four datasets, all $> 0$, and demonstrate consistent collaborative gains.

- *trivial baselines*: 1) *Best Single* expert, essentially $\arg\max_{\mathbf{x}} f(\mathbf{x})$ for $\mathbf{x} \in \{\mathbf{x}_i\}_{i=1}^n$; 2) *Prediction Merge*, where the predictions of $\{\mathbf{x}_i\}_{i=1}^n$ are ensembled via plurality vote (if applicable).

- *static weight*: these approaches conduct model merging independent of the task and utility function $f$: *Data Merge*, *Uniform Soup* [97], and *Dare-Ties* [108, 103].

- *dynamic weight*: these approaches optimize model weights based on the utility function $f$: *Greedy Soup* [97], *Pack of LLMs* [66], *LoraHub* [40], and *Model Swarms* [26].

- *static role*: we employ two structures in hand-crafted agent systems [52]: *chain* and *star*.

- *dynamic role*: these approaches optimize model roles and connections based on $f$: *GPT-Swarm* [120], *Meta-Agent* [37], *Agent-Prune* [109], *GNNs* [110], *AgentVerse* [11], and *MACNet* [73].

**Data and Evaluation** We compare HETEROGENEOUS SWARMS against baselines on 12 datasets spanning 4 categories: 1) knowledge: MMLU-pro [94], Knowledge Crosswords [19], and COM2 [23]; 2) reasoning: GSM8k [13], NLGraph [89], and Normad [76]; 3) agent: GAIA-text [67], the knowledge graph and lateral thinking puzzle subtasks of AgentBench [61]; 4) miscellaneous: long-context with Qasper [16], reliability with AbstainQA [25], and LLM-as-a-judge with WoW [106].

# 5 Results

We present the performance of HETEROGENEOUS SWARMS and baselines on the 12 tasks in Table 1.

**HETEROGENEOUS SWARMS consistently discovers state-of-the-art multi-LLM systems.** HETEROGENEOUS SWARMS achieves the best performance on 11 of the 12 datasets, outperforming the second-best approach by 18.5% on average. This indicates that starting from the same pool of initial LLMs, HETEROGENEOUS SWARMS could flexibly discover multi-LLM systems that adapt to diverse tasks and contexts spanning knowledge, reasoning, and more.

**Role and weight are disproportionately important for different tasks.** For knowledge tasks, weight baselines (STATIC WEIGHT and DYNAMIC WEIGHT) outperform role baselines (STATIC ROLE and DYNAMIC ROLE) by 4.3% on average. However, for agent tasks, role baselines are 9.2% better. This indicates that role and weight have varying importance in different tasks: by jointly optimizing roles and weights in multi-LLM systems, HETEROGENEOUS SWARMS flexibly adapts to both scenarios. We further investigate their importance in Section 6.

**Dynamic adaptation works better than static engineering.** We find that DYNAMIC WEIGHT approaches outperform STATIC WEIGHT by 30.1% across tasks, while DYNAMIC ROLE outperforms STATIC ROLE by up to 8.2%. This indicates that instead of hand-crafted ways to design roles or merge weights, dynamic adaptation approaches guided by utility function $f$ offer better adaptation. HETEROGENEOUS SWARMS employs role-step and weight-step to dynamically adapt both for $f$, resulting in flexible multi-LLM systems adapted to diverse tasks and applications.

# 6 Analysis

**Collaborative Gains** To justify the cost of calling multiple LLMs in topological order, a multi-LLM system should unlock $1 + 1 > 2$ effects: the multi-LLM collaboration should produce *collaborative gains* compared to employing a single LLM. Concretely:

> For problem $q$, if $p\%$ of the component LLMs could solve it individually, then the multi-LLM system should have a larger than $p\%$ likelihood of solving it.

To quantify this, we group problems in dataset $\mathcal{D}$ by how many component LLMs could solve it individually: problem $q \in B_n$ if $n$ of the $N$ component LLMs could solve it. We then calculate the metric *Collaborative Gain* as:

$$\text{C-Gain} = \sum_{n=1}^{N} \frac{|B_n|}{|\mathcal{D}|} \big( \text{Acc}(B_n) - \text{EA}(B_n) \big)$$

where $\text{Acc}(B_n)$ denotes the accuracy of the multi-LLM systems for problems in $B_n$, $\text{EA}(B_n)$ denotes the Expected Accuracy $n/N$, $|B_n|$ and $|\mathcal{D}|$ denote the number of problems in the bucket and the dataset. If a multi-LLM system satisfies the principle, then $\text{Acc}(B_n) > \text{EA}(B_n)$ and $\text{C-Gain} > 0$.

| | Knowledge | | | Reasoning | | | Agent | | | Miscellaneous | | |
|---|---|---|---|---|---|---|---|---|---|---|---|---|
| | MMLU-pro | K-Cross | COM2 | GSM8k | NLGraph | Normad | GAIA-text | AB-kg | AB-ltp | Qasper | AbstainQA | WoW |
| Role Baselines | 0.218 | 0.318 | 0.486 | 0.323 | 0.531 | 0.480 | 0.095 | 0.318 | 0.132 | 0.205 | 0.042 | 0.488 |
| Weight Baselines | 0.222 | 0.352 | 0.490 | 0.325 | 0.538 | 0.494 | 0.082 | 0.342 | 0.112 | 0.169 | 0.067 | 0.516 |
| Ours w/o Role | 0.242 | 0.352 | 0.515 | 0.392 | 0.530 | 0.564 | 0.107 | 0.317 | 0.140 | 0.222 | 0.133 | 0.539 |
| Ours w/o Weight | 0.237 | 0.342 | 0.492 | 0.363 | 0.588 | 0.557 | 0.143 | 0.325 | 0.164 | 0.241 | 0.119 | 0.510 |
| Ours Full | 0.312 | 0.450 | 0.579 | 0.481 | 0.660 | 0.588 | 0.250 | 0.425 | 0.215 | 0.266 | 0.220 | 0.590 |
| Consistent? | TRUE | TRUE | TRUE | TRUE | FALSE | TRUE | TRUE | FALSE | TRUE | TRUE | TRUE | TRUE |

Table 2: Ablation study of removing the role-step or weight-step in HETEROGENEOUS SWARMS, comparing whether the importance of role/weight is consistent between baselines and our approach. The pattern is consistent in 10 out of 12 datasets, confirming that model roles and weights could have different levels of importance for varying tasks.

We present the collaborative gains of HETEROGENEOUS SWARMS in Figure 3. HETEROGENEOUS SWARMS achieves consistent positive collaborative gains with an average of 0.213. For problems

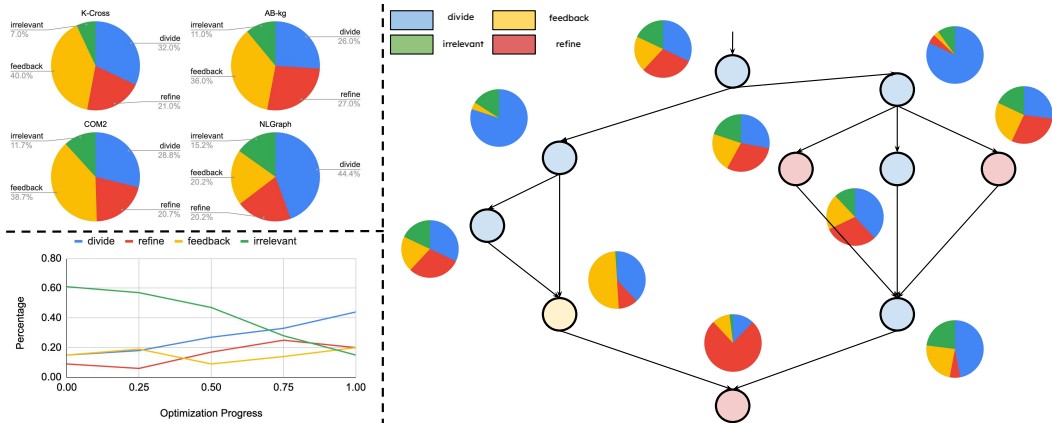

Figure 4: Analyzing the roles in Multi-LLM systems. Top left: the percentage of LLM roles aggregated per dataset. Bottom left: the change of LLM roles in the optimization process for NLGraph. Right: Per-LLM role distribution in the best-found multi-LLM system for NLGraph. Together these figures demonstrate the heterogeneous roles in the multi-LLM systems by HETEROGENEOUS SWARMS.

in bucket $B_0$, i.e. none of the initial LLMs could solve individually, HETEROGENEOUS SWARMS discovers multi-LLM systems that solve 18.1% of them on average. We additionally find that HETEROGENEOUS SWARMS has greater C-Gain than baseline in Table 12. This indicates that HETEROGENEOUS SWARMS could find adapted multi-LLM systems with new compositional skills and substantial collaborative gains.

**Importance of Role and Weight through Ablation Study**   In section 5, we discover that model roles and weights could be disproportionately important for different tasks based on baseline performance. We compare the trend with our approach, specifically through disabling either the role step or the weight step in HETEROGENEOUS SWARMS (*w/o Role* and *w/o Weight*). Let $B_r$ and $B_w$ denote the average performance of role/weight-based baselines, then the importance of role/weight is consistent when the following logic expression is True:

$$\big((\text{\textit{w/o Role}} < \text{\textit{w/o Weight}}) \text{ AND } (B_r > B_w)\big)$$
$$\text{OR } \big((\text{\textit{w/o Role}} > \text{\textit{w/o Weight}}) \text{ AND } (B_r < B_w)\big)$$

We present performance of the ablated settings and the value of the logic expression across 12 datasets in Table 2. Results demonstrate that roles and weights could have varying importance (e.g. weight is more important for knowledge tasks while role is more important for agent) and such importance is consistent in 10 of the 12 tasks. In addition to jointly adapting model roles and weights, HETEROGENEOUS SWARMS offers insights into their importance for the task at hand.

**Role Statistics**   We manually examine the input/output of multi-LLM systems, identifying four potential roles of individual models in the multi-LLM system: 1) *divide*, where the LLM identifies and solves part of the problem; 2) *refine*, where the LLM proposes a new (sub)answer based on previous steps; 3) *feedback*, where the LLM provides feedback on previous steps; 4) *irrelevant*, where the LLM fails to generate relevant text. We employ Gemini-as-a-judge [87] (GEMINI-1.5-FLASH) to automatically identify the role of individual LLMs and conduct three analysis in Figure 4.

We analyze the variation of model roles across tasks in Fig.4, top left. For the graph reasoning task NLGraph, there is greater *divide* and conquer to solve part of the problem; for the knowledge tasks such as K-Crosswords, there is greater *feedback* to identify knowledge gaps in existing answers. Guided by different $f$s, HETEROGENEOUS SWARMS discover multi-LLM systems with different role distributions.

We investigate the roles of LLMs in different positions of the DAG on NLGraph in Fig.4, right. We find that individual LLMs do have heterogeneous role distributions given their topological position: for the branching nodes there is often higher *divide*, while for the converging nodes there is often higher *refine* and *feedback*. This indicates that HETEROGENEOUS SWARMS successfully discovers multi-LLM systems where individual LLMs play heterogeneous roles.

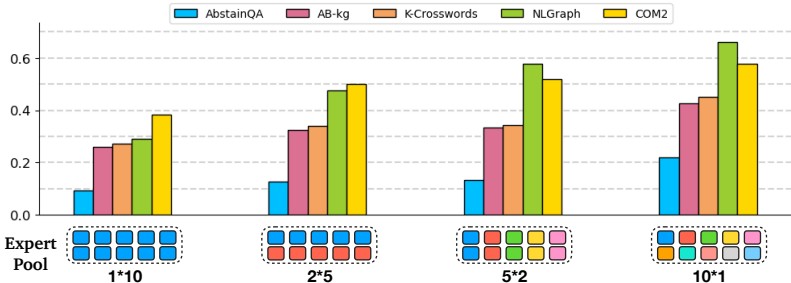

Figure 5: HETEROGENEOUS SWARMS with increasing levels of diversity in initial LLMs. Results show a general upward trend and an 89% increase on average from the least to most diverse models.

We plot the change of model roles in the optimization process in Figure 4, bottom left, when adapting to NL-Graph. We observe that *irrelevant* roles gradually decrease while the three other functional roles consistently increase in the course of optimization. This indicates that by integrating weight optimization, HETEROGENEOUS SWARMS improves the quality of LLMs to adapt to the task at hand.

|  | K-Cross | | NLGraph | | AB-kg | |
|---|---|---|---|---|---|---|
|  | Acc | Speedup | Acc | Speedup | Acc | Speedup |
| original | 0.450 | 0.0% | 0.660 | 0.0% | 0.425 | 0.0% |
| $\tau = 0.05$ | 0.438 | 3.8% | 0.628 | 4.9% | 0.392 | 9.2% |
| $\tau = 0.1$ | 0.392 | 12.4% | 0.609 | 9.3% | 0.375 | 19.8% |
| $\tau = 0.2$ | 0.352 | 36.1% | 0.581 | 13.6% | 0.383 | 21.4% |
| $\lambda = 0.01$ | 0.436 | 1.2% | 0.642 | 0.9% | 0.400 | 3.4% |
| $\lambda = 0.05$ | 0.425 | 3.5% | 0.614 | 2.3% | 0.383 | 9.1% |
| $\lambda = 0.1$ | 0.426 | 7.1% | 0.598 | 2.5% | 0.392 | 12.8% |

Table 3: Encouraging sparsity in multi-LLM systems with thresholed pruning ($\tau$) or normalization ($\lambda$). These strategies bring various tradeoffs bewteen performance and inference speedup.

**Encouraging Sparsity** We observe that if the network is dense, i.e. when one LLM takes the output of too many LLMs as input, it might lead to irrelevant outputs. In addition, encouraging sparsity in the network would also reduce context lengths and hence inference costs. To this end, we encourage sparsity in the multi-LLM network by two means: 1) by setting a threshold $\tau \in [0, 1]$ and pruning the continuous adjacency $\mathcal{A}$ with $\mathcal{A}' = \{a_{ij} \cdot \mathbb{1}[\max(a_{ij} - \tau, 0)]\}$; 2) by adding a $\mathcal{L}_1$ normalization term to the utility function: $f' = f + \lambda \|\mathcal{A}\|_1$. We employ various values of $\tau$ and $\lambda$: Table 3 presents the performance and speedup calculated by the percentage of reduced connections in the network. Results demonstrate that $\mathcal{L}_1$ better retains performance while thresholding presents a larger reduction in inference cost.

**Diversity Matters** HETEROGENEOUS SWARMS operates on the assumption that the multiple large language models would complement each other based on their diverse expertise. To investigate whether this model diversity is crucial, we conduct an controlled experiment where we fix the LLM pool size as 10, but with $a$ distinct models each repeated $b$ times. We denote this as $a \times b$ and employ $1 \times 10$, $2 \times 5$, $5 \times 2$, and $10 \times 1$ (default setting) with increasing diversity. Figure 5 illustrates that the diversity of initial LLMs does help greatly, with an average relative improvement of 89% from the least to most diverse.

# 7 Related Work

A growing line of research focuses on *multi-LLM collaboration*, where multiple models collaboate and complement each other. These approaches focus on either model roles or weights.

Role-based approaches typically rely on assigning roles to LLMs through prompt engineering [21, 25]. Multiple LLMs, or even just a single LLM seeded with different prompts, then collaborate with their prompt-induced roles through exchanging generated texts. For example, specialized LLMs could augment a general-purpose model [24, 79]; LLMs could generate feedback for each other's responses to collectively self-refine [25, 7]; (multi-)agent systems could divide and conquer complex problems [98, 33, 91, 60, 118, 71, 47, 65, 55, 99, 43, 56, 92, 86, 53, 75, 68, 9, 17, 119]; multiple LLMs could debate and compete with each other to find better answers [57, 21]. These approaches are often hindered by the need for prompt engineering and the effectiveness of prompts for model steerability [83, 77], thus HETEROGENEOUS SWARMS uniquely interprets model roles as input-output relationships, optimizing directed acyclic graphs of LLMs to learn contextual roles and specialization.

Weight-based approaches typically focus on adapting the logits/weights of multiple LLMs, notably through mixture-of-experts or model merging [102]. The hidden states or logit distributions of multiple models could be selected, routed, and aggregated based on various MoE mechanisms [54, 30]. In addition, static [108, 103, 45] and dynamic [66, 1, 40] model merging approaches incorporate the diverse expertise of heterogeneous LLMs into a single model in zero-shot and adaptation settings. We continue to believe that weight adaptation is crucial for specializing individual LLMs in multi-LLM systems: guided by the first successes of evolutionary algorithms in weight-based collaboration [26] and LLMs in general [1, 27, 32], HETEROGENEOUS SWARMS employs swarm intelligence to optimize model weights guided by each LLM's individual contribution to the multi-LLM system.

HETEROGENEOUS SWARMS uniquely offers a flexible methodology to jointly optimize the roles and weights of diverse LLMs, discovering and adapting novel multi-LLM systems.

## 8    Conclusion

We propose HETEROGENEOUS SWARMS, an algorithm to jointly optimize model roles and weights for multi-LLM systems and collaborative generation. By rotating between role-step and weight-step to optimize the network of LLMs as well as model weights, HETEROGENEOUS SWARMS discovers directed acyclic graphs of LLMs that could be called in topological order to adapt to a given task. HETEROGENEOUS SWARMS outperforms 17 role and weight-based baselines on 12 tasks, demonstrating collaborative gains and heterogeneous roles through multi-LLM collaboration.

## Acknowledgements

This research was developed with funding from the Defense Advanced Research Projects Agency's (DARPA) SciFy program (Agreement No. HR00112520300). The views expressed are those of the author and do not reflect the official policy or position of the Department of Defense or the U.S. Government. This material is based upon work supported by the Defense Advanced Research Projects Agency and the Air Force Research Laboratory, contract number(s): FA8650-23-C-7316. Any opinions, findings and conclusions, or recommendations expressed in this material are those of the author(s) and do not necessarily reflect the views of AFRL or DARPA. Shangbin Feng would like to thank the support of the IBM PhD Fellowship and Jane Street Graduate Research Fellowship.

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

## A    Limitations

HETEROGENEOUS SWARMS by default operates with a pool of LLMs sharing the same model architecture: this is to ensure that models share the same parameter space for weight-step optimization. If the pool of LLMs contain different architectures, then HETEROGENEOUS SWARMS could either operate with only the role-step to optimize the multi-LLM network, or employ token-level collaboration for the weight step similar to Liu et al. [58] and Feng et al. [26].

Both the optimization and inference process of HETEROGENEOUS SWARMS features calling multiple LLMs, increasing its potential cost. We propose various strategies to alleviate this: by encouraging sparsity in the multi-LLM network in Table 3, by analyzing the time complexity in Section C, and by employing Dropout-W/R strategies in Figure 9, we show that HETEROGENEOUS SWARMS could speed up with one or multiple strategies used in conjunction.

While we conduct experiments where all the nodes in the network are neural language models, we highlight several other possibilities we weren't able to evaluate: employing black-box models in the LLM pool, which is compatible if we skip these models in the weight-step; employing tools and APIs as nodes in more agentic setups [59, 28, 39, 78, 73, 113, 69, 74, 20, 105, 90, 112, 64, 22, 8, 111, 85, 62, 15, 41], as long as they define an input-output relationship.

We envision two extra steps to take to further expand the expressive power of HETEROGENEOUS SWARMS: allowing for cycles in the multi-LLM network in case one model needs to be called multiple times, together with an exit function to decide when to pull out from the cycle; allowing for instance-level changes to the multi-LLM network with a learnable edit function.

## B    Ethics Statement

Since multiple LLMs are called in topological order of the multi-LLM system, we envision potential risks when one or several of the component LLMs are unintentionally or intentionally compromised. For example, if one of the LLMs were to exhibit substantial social biases [51] or malicious intent [117], it might pass on to future models in the network and result in cascading effects. There are further risks in agentic [34, 104, 96, 72, 18, 31, 70, 115, 4, 10, 12, 14, 38, 42, 63, 81, 93, 101, 107, 114, 84, 2] versions of HETEROGENEOUS SWARMS where one model could be biased or malicious. We thus argue that HETEROGENEOUS SWARMS should be initialized with carefully selected pool of specialized LLMs and envision future work on identifying harmful/malicious components in multi-LLM collaboration.

## C    Analysis (cont.)

**Scaling Multi-LLM Collaboration**    Recent research explored the inference-time scaling of a single, often gargantuan, LLM [82, 5, 100]. However, it is unclear whether inference-time scaling is possible for small language models, while we believe HETEROGENEOUS SWARMS offers a way of scaling through topological message passing and collaborative generation. We conduct a scaling study by employing 2, 4, 6, 8, and 10 (default) of initial LLM checkpoints as the starting model swarn and run HETEROGENEOUS SWARMS: results in Figure 6 demonstrates that HETEROGENEOUS SWARMS indeed offers an inference-time scaling paradigm for smaller language models through multi-LLM collaboration.

**Time Complexity**    Given a pool of $n$ LLMs, we employ a swarm of $N$ graphs and $M$ model assignment instances. We measure the complexity by the number of model inferences as they are the main computational cost. At optimization time, the role step calls model inference $O(nN)$ times and the weight step $O(nM)$ times, so the overall optimization cost per iteration is $O(n(N+M)|f|)$ where $|f|$ denotes the adaptation data size, linear to $n$, $N$, and $M$. At inference time, the cost is $O(n)$ where the adapted LLMs are called in topological order of the network. We further illustrate the empirical complexity by plotting the time per

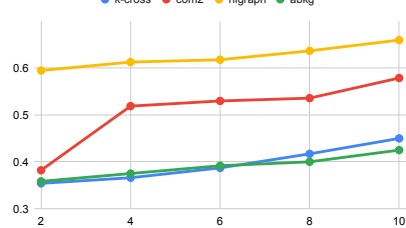

Figure 6: Scaling the number of LLM experts from 2 to 10 results in consistent improvements across 4 datasets.

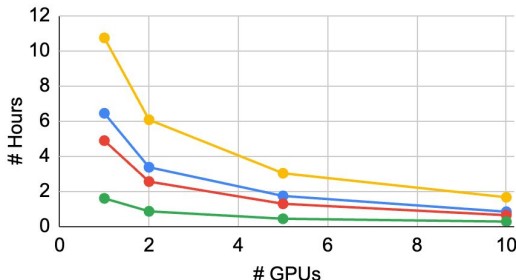

Figure 7: Optimization time changes with the number of A100 40GB GPUs employed. We employ 1, 2, 5, and 10 GPUs since they could be divided by 10, the amount of LLMs.

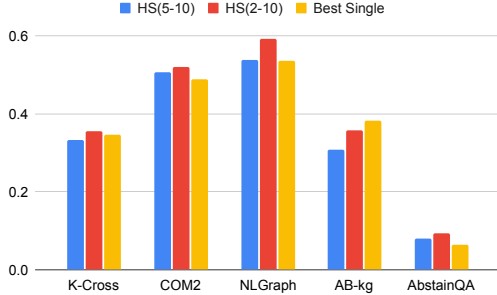

Figure 8: The collaboration of the all-but-top-1 and bottom-half LLMs through HETEROGENEOUS SWARMS could beat the top-1 individual LLM.

iteration given different amount of A100 40GB GPUs in Figure 7. We present the optimization and inference cost of approaches in Table C: HETEROGENEOUS SWARMS is more expensive at optimization time and on-par with most baselines at inference time, while the performance improvements and collaborative gains justify the additional optimization cost.

**HS(weak) > strong**   A successful collaboration of multiple weak models can beat a stronger LLM: we withhold the top-1 LLM in the given task and discover a multi-LLM system out of the remaining 9 models with HETEROGENEOUS SWARMS (HS(2-10)). Additionally, we also withhold the top half and evaluate the collaboration of the bottom half (HS(6-10)). Figure 8 demonstrates that HETEROGENEOUS SWARMS enables the collaboration of weaker models to outperform the top-1 individual LLM, enabling the weak-to-strong transition through collaborative generation.

| Approach | optimization | inference |
|---|---|---|
| best single | / | 1 |
| pred. merge | / | n |
| static weight | / | 1 |
| dynamic weight | n | 1 |
| static role | / | n |
| dynamic role | nN | n |
| ours | n(N+M) | n |

Table 4: Optimization and inference cost of different types of approaches. $n$ is the number of models, $N$ is the number of graphs, and $M$ is the number of graph-model assignments.

**Accelerating with Dropout-R/W**   To further speedup the optimization process, we process to randomly skip the role-step or weight-step with $d_r\%$ or $d_w\%$ likelihood. We illustrate the performance with $d_r, d_w \in \{0.2, 0.5, 0.8\}$ in Figure 9. It is illustrated that the acceleration only comes with a minor performance drop, showing the flexibility of HETEROGENEOUS SWARMS to adapt to different computational budget.

**Another LLM**   We by default employ GEMMA-7B as the component LLMs. We additionally employ MISTRAL-7B ((*mistralai/Mistral-7B-Instruct-v0.3*)) [46] for HETEROGENEOUS SWARMS and present performance in Table 5. Results show that HETEROGENEOUS SWARMS is compatible with MISTRAL-7B too, outperforming the best single Mistral-based LLM.

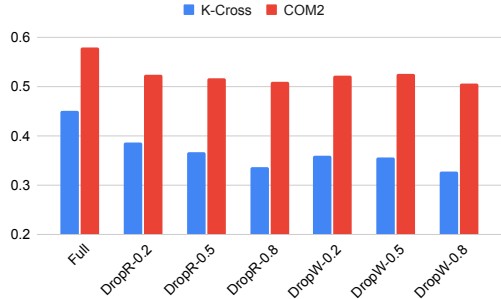

Figure 9: By randomly skipping the role-step and weight-step through Drop-R and Drop-W, we could speed up the optimization process.

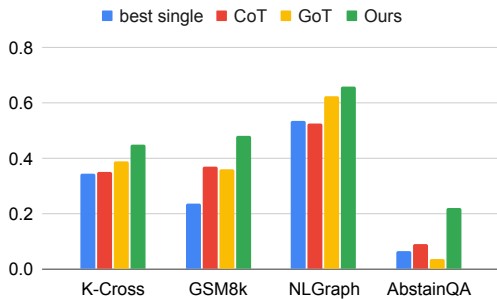

Figure 11: Our approaches outperform a single model empowered with reasoning enhancement approaches.

**Comparison with a Larger Model** We compare the collaboration of 3 *gemma-2-9b* models against a single *gemma-2-27b* model, as well as the collaboration of 6 *gemma-3-4b* models against a single *gemma-3-27b* model in Figure 10. It is demonstrated that through HETEROGENEOUS SWARMS, the multi-LLM systems of smaller models could outperform larger models.

| Setting | MMLU-pro | K-Cross | GSM8k | NLGraph | AbstainQA |
|---|---|---|---|---|---|
| Best Single | 0.146 | 0.364 | 0.303 | 0.325 | 0.081 |
| Ours | 0.212 | 0.417 | 0.313 | 0.505 | 0.123 |

Table 5: Performance with 10 Mistral LMs.

**Comparison with Enhanced Reasoning Approaches** We compare our approach against chain-of-thought [95] and graph-of-thought [3] with the best initial LLM expert in Figure 11. The performance improvements suggest that by having multiple models collaborate, their expertise could complement each other and achieve more.

**Generalization to Unseen Tasks** We optimize multi-LLM systems based on the Knowledge Crosswords dataset and evaluate them on WikiDYK [?], two datasets in the encyclopedic knowledge domain, in Table C. Results demonstrate that our approach better generalizes to unseen tasks.

**Collaborative Gains of Baselines** We compare the collaborative gains of our approach with two model merging baselines in Figure 12. It is demonstrated that HETEROGENEOUS SWARMS does yield higher collaboration gains in the optimized multi-LLM systems.

**Collaboration of Different Model Sizes** We run the role-step of HETEROGENEOUS SWARMS with different

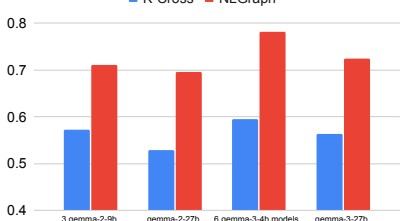

Figure 10: Multi-LLM systems of multiple smaller models could outperform a single larger model, with similar amounts of total parameters.

| Approach | K-Cross | WikiDYK |
|---|---|---|
| greedy soup | 0.355 | 0.525 |
| pack of llms | 0.352 | 0.513 |
| lorahub | 0.291 | 0.501 |
| model swarms | 0.428 | 0.527 |
| gpt-swarm | 0.320 | 0.472 |
| meta-agent | 0.276 | 0.475 |
| agent-prune | 0.321 | 0.431 |
| gnns | 0.339 | 0.365 |
| ours | 0.450 | 0.566 |

Table 6: Generalization to unseen tasks: optimizing on K-Cross and evaluating on the held-out task of WikiDYK.

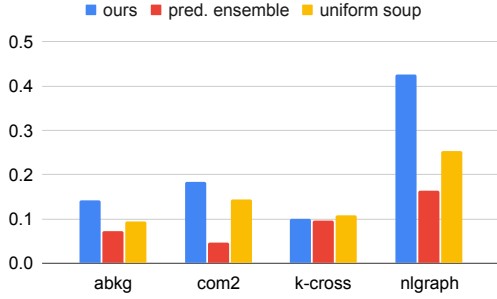

Figure 12: HETEROGENEOUS SWARMS achieves higher collaborative gains compared to baselines.

| Setting | K-Cross | GSM8k | NLGraph | AbstainQA |
|---|---|---|---|---|
| full 7b | 0.450 | 0.481 | 0.660 | 0.220 |
| half 7b half 2b | 0.318 | 0.371 | 0.492 | 0.094 |
| full 2b | 0.265 | 0.215 | 0.392 | 0.082 |
| single 7b | 0.346 | 0.237 | 0.535 | 0.065 |
| single 2b | 0.194 | 0.148 | 0.364 | 0.023 |

Table 7: Collaboration of mixed-size models, compared to fixed-size and best single models.

model size settings, with *Gemma-7b* and *Gemma-2b* models. 1) full 7b (default), 2) half 7b half 2b, 3) full 2b, 4) best single 7b, 5) best single 2b. We present results in Table C: half 7b half 2b outperforms both best single 7b and full 2b indicates that mixed-size collaboration works under HETEROGENEOUS SWARMS as well.

## D  Experiment Details

**Dataset Details**   We employ 12 datasets to evaluate HETEROGENEOUS SWARMS and baselines spanning knowledge, reasoning, agent, and miscellaneous capabilities. We filter examples in GAIA to retain examples where the human-provided tool use contexts could support the final answer for GAIA-text. We incorporate each tool use context for the input of one LLM in the multi-LLM network respectively. We truncate the context in Qasper in ten-fold and incorporate each chunk as context for an individual LLM. We by default sample 200 examples for optimization and 1,000 for evaluation, while downsampling if there's not enough data. We present data statistics in Table 8. MMLU-pro, Knowledge Crosswords, COM2, Normad, AgentBench-KG, AbstainQA, and WoW are evaluated in multiple-choice settings. GSM8k, NLGraph, and GAIA-text are evaluated via exact match. AgentBench-LTP and Qasper are evaluated by Gemini for answer similarity on a scale of 1 to 10, normalized to 0 to 1. We also employ the z-test with the one-tailed hypothesis and present statistical significance test results on the datasets.

| Dataset | Source | Size | |
|---|---|---|---|
| | | dev | test |
| MMLU-pro*** | Wang et al. [94] | 70 | 1000 |
| K-Crosswords*** | Ding et al. [19] | 200 | 1000 |
| COM$^2$** | Fang et al. [23] | 317 | 1000 |
| GSM8k | Cobbe et al. [13] | 200 | 1000 |
| NLGraph | Wang et al. [89] | 200 | 1000 |
| Normad** | Rao et al. [76] | 500 | 2000 |
| GAIA-text | Mialon et al. [67] | 13 | 28 |
| AgentBench-KG | Liu et al. [61] | 50 | 120 |
| AgentBench-LTP | Liu et al. [61] | 20 | 50 |
| Qasper | Dasigi et al. [16] | 100 | 100 |
| AbstainQA*** | Feng et al. [25] | 200 | 1000 |
| WoW | Yao et al. [106] | 200 | 1000 |

Table 8: Statistics of employed datasets. *, **, and *** indicates the improvement on this dataset is statistically significant with $p < 0.1$, $p < 0.05$, and $p < 0.01$ with one-tailed z-test.

**Implementation Details**  We employ the 10 LLM experts in Feng et al. [26] as the initial swarm of models. We employ the Gemma-based [29] versions for experiments in the main paper for a fair comparison and employ the Mistral-based [46] versions for experiments in Table 5. Aside from the hyperparmaters specified in Section 4, we run grid search over other hyperparameters and report the best-found expert based on utility function $f$. Specifically, $\phi_v \in \{0.1, 0.2, 0.3\}$, $\phi_p \in \{0.1, 0.2, 0.3, 0.4, 0.5\}$, $\phi_g \in \{0.2, 0.3, 0.4, 0.5, 0.6\}$, $\phi_w \in \{0.01, 0.05, 0.1\}$, $\lambda \in \{0.5, 0.6, 0.7, 0.8, 0.9, 1.0\}$. We run up to 50 to 200 runs by randomly choosing over these hyperparameter search settings and report the best-found expert on utility function $f$. Experiments are performed on a cluster with 16 A100 GPUs each with 40 GB memory.

**Baseline Details**  For *best single*, *prediction merge*, *data merge*, *uniform soup*, *dare-ties*, *greedy soup*, *pack of LLMs*, *lorahub*, and *model swarms*, we follow the settings in Feng et al. [26]. For *chain*, we organize the 10 initial LLMs into a chain, randomly permute their assignments for 20 times, and report the best-found chain on $f$. For *star*, we organize them into a 1-8-1 structure, randomly permute their assignments for 20 times, and report the best-found star on $f$. For *gpt-swarm*, we employ its method to optimize the network structure and do not consider prompt optimization for a fair comparison. To make the optimization compatible to our non-differentiable setting, we modify it into interpolation of adjacency matrices based on $f$ utility values. For *meta-agent*, we employ Gemini-pro (*gemini-pro-1.5-001*) with the original prompt to iteratively suggest multi-LLM structures, while representing each individual LLM by a natural language description of their training data and expertise. For *agent-prune*, we randomly sample multi-LLM structures, conduct pruning and re-evaluate, repeat this process for 20 times and report the best-found structure on $f$ across runs and pruning degrees. For *GNNs*, we employ graph attention networks to encode the multi-LLM network with randomly initialized node features, and learn a link prediction task based on whether adding this edge between two LLMs would lead to a performance increase on $f$ or not. At inference-time, the GNN conducts link prediction for $n \times n$ edges to obtain a multi-LLM network.

**Prompts**  We present the prompts employed in various contexts in Tables 9, 10, and 11. We make dataset-specific changes to the general prompt format if necessary.

**Qualitative Examples**  We present two working examples of the multi-LLM systems in HETEROGENEOUS SWARMS, on Knowledge Crosswords [19] and NLGraph [89] datasets in Figures 13 and 14. These examples demonstrate that HETEROGENEOUS SWARMS discovers multi-LLM systems to adapt to the task and play heterogeneous roles through collaborative generation.

**Illustrating the Optimization**  We illustrate the changes in the best $f$ utility values for the swarm of graphs and swarm of LLMs in Figure 15: it is illustrated that there is co-improvement of roles and weights thanks to the alternating optimization algorithm.

> *prompt for the first/entry-point LLM*
"Please answer the following question."

> *prompt for the middle/intermediate LLMs*
"Please answer the following question with the help of previous responses, feel free to ignore wrong or unhelpful responses."

> *prompt for the final/end-point LLM*
"Please answer the following question with the help of previous responses, feel free to ignore wrong or unhelpful responses. Make sure to provide a final and definitive answer."

Table 9: Default LLM instructions in the multi-LLM system.

"Please evaluate how similar is the following response to the ground truth answer. Please rate the response on a scale of 1 to 10, where 1 is the worst and 10 is the best. Please respond with Rating: ?/10 first and then provide your reason.

Ground truth: [ground truth]
Generated response: [generated response]"

Table 10: Default Gemini-as-a-judge evaluation prompt.

"Below is the output of a model solving part of a problem. Please judge whether the output is one of the four scenarios: 1. solving part of the problem, 2. refining the previous answer, 3. providing feedback, 4. irrelevant.

The problem is: [problem text]
The output is: [intermediate output]"

Table 11: Prompt for Gemini to evaluate the roles of intermediate model outputs in multi-LLM systems.

**Algorithm 4:** DAG Decoder (G-decode)

---

**Input:** continuous adjacency matrix $\mathbf{A} \in \mathbb{R}^{n \times n}$ where $a_{ij}$ denotes the likelihood of a directed edge from model $\mathbf{x}_i$ to model $\mathbf{x}_j$

edges $\mathcal{G} = \varnothing$, remaining node set $\mathcal{R} = \{1, \cdots, n\}$, existing node set $\mathcal{E} = \varnothing$

select end node based on inverse out degrees $k = \text{top-p-sampling}(\{1/\sum_{j=1}^{n} a_{ij}\}_{i=1}^{n})$

$\mathcal{R}.\text{remove}(k), \mathcal{E}.\text{add}(k)$

**while** $\mathcal{R} \neq \varnothing$ **do**

   select remaining node based on out degrees $u = \text{top-p-sampling}(\{\sum_{j=1}^{n} a_{ij}\}_{i \in \mathcal{R}})$

   $\mathcal{G}.\text{add}(\{u \to v\})$ with prob. $\frac{\exp(a_{uv})}{\sum_{i \in \mathcal{E}} \exp(a_{ui})}, \forall v \in \mathcal{E}$

   $\mathcal{R}.\text{remove}(u), \mathcal{E}.\text{add}(u)$

**end**

**return** directed acyclic graph $\mathcal{G}$, defines an input-output mapping based on $\{\mathbf{x}_i\}_{i=1}^{n}$

---

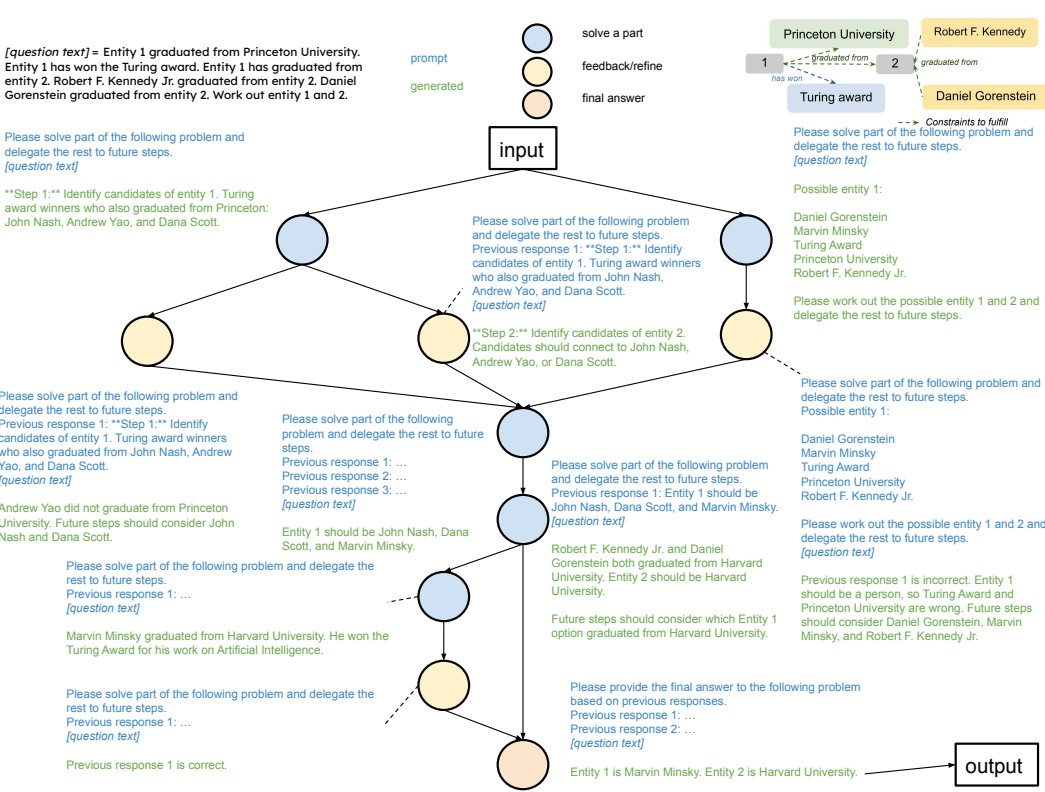

Figure 13: Working example one of HETEROGENEOUS SWARMS.

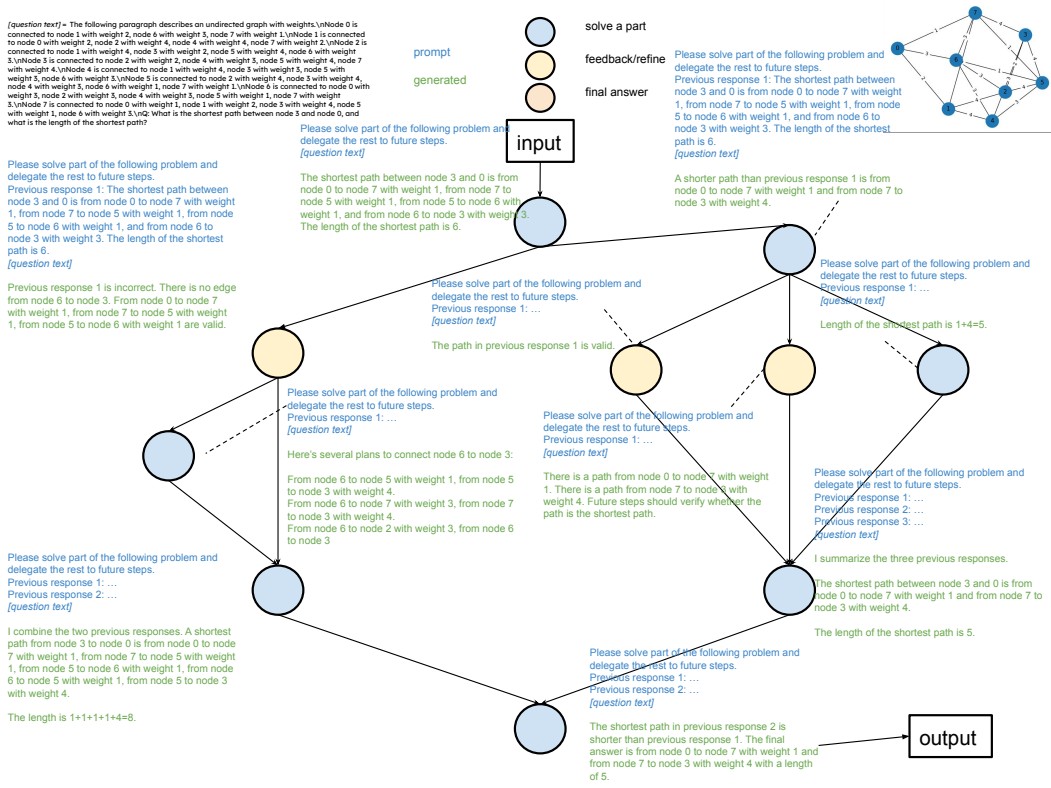

Figure 14: Working example two of HETEROGENEOUS SWARMS.

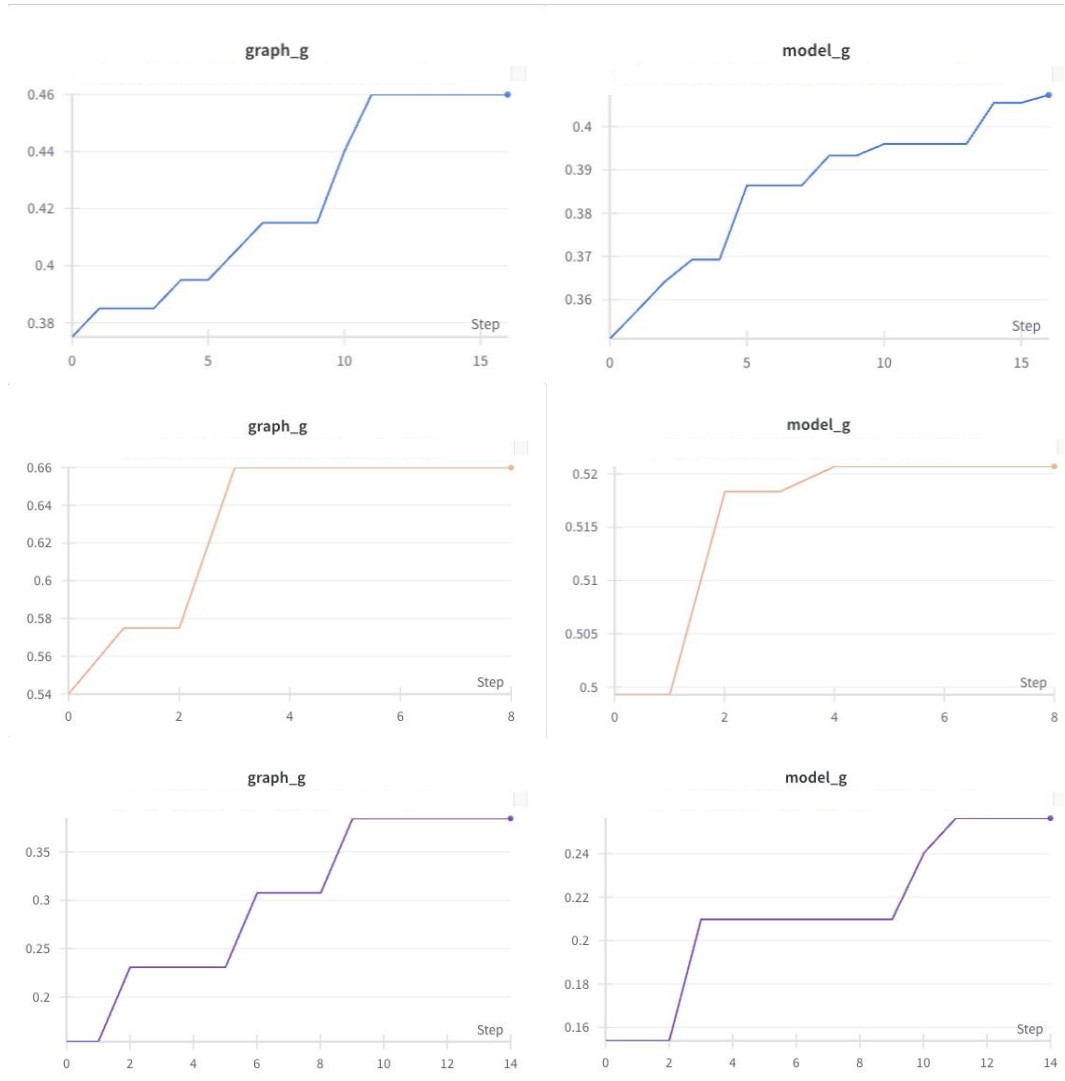

Figure 15: Illustrating how the utility of the role-step (left) and weight-step (right) changes through time on three datasets, Knowledge Crosswords, NLGraph, and AB-kg. We observe co-improvement in the utility of graphs and model weights.

