# OpenReview forum: "Heterogeneous Swarms: Jointly Optimizing Model Roles and Weights for Multi-LLM Systems"
_NeurIPS.cc/2025/Conference — NeurIPS 2025 poster_

### Official Review · Reviewer_cehQ · 2025-06-21

**Clarity:** 3
**Significance:** 3
**Originality:** 2
**Rating:** 5
**Confidence:** 4

**Summary:**

This paper proposes Heterogeneous Swarms, a method for jointly optimizing model roles and weights in multi-LLM systems. The approach represents multi-LLM systems as directed acyclic graphs (DAGs) and uses particle swarm optimization (PSO) to iteratively optimize both the graph structure (role-step) and model weights (weight-step). The method introduces G-DECODE for converting continuous adjacency matrices to discrete DAGs and JFK-score for evaluating individual model contributions. Experiments across 12 tasks show an 18.5% average improvement over 17 baselines.

**Questions:**

1. **Theoretical Justification**: Can you provide theoretical analysis of when and why PSO should work well for this problem? What are the convergence guarantees?
2. **Scalability Analysis**: How does performance scale with the number of expert models (beyond 10)? What are the practical limits of the approach?
3. **Architecture Heterogeneity**: How could the method be extended to handle expert models with different architectures? This seems crucial for practical deployment.
4. **Hyperparameter Robustness**: How sensitive is the method to PSO hyperparameters? Could you show performance across a range of settings rather than just the best-found configuration?
5. **Computational Efficiency**: Beyond the proposed speedup strategies, are there more fundamental ways to reduce the computational overhead while maintaining performance?

**Ethical Concerns:**

["NO or VERY MINOR ethics concerns only"]

**Final Justification:**

The authors' rebuttal has satisfactorily resolved my concerns regarding scalability and hyper-parameter sensitivity, making the work feel more robust. Consequently, I am raising my score from 4 to 5.

**Limitations:**

yes

**Quality:**

3

**Strengths And Weaknesses:**

### Strengths

1. **Novel Joint Optimization Approach**: The paper addresses an important limitation in existing multi-LLM systems by jointly optimizing both roles (graph structure) and weights, rather than treating them as separate problems.
2. **Comprehensive Experimental Evaluation**: The evaluation spans 12 diverse datasets across knowledge, reasoning, agent, and miscellaneous tasks, with comparisons against 17 baselines across multiple categories.
3. **Practical Algorithmic Contributions**:
    - G-DECODE provides a principled way to convert continuous optimization to discrete DAG structures
    - JFK-score offers a novel metric for quantifying individual model contributions in multi-LLM systems
4. **Thorough Analysis**: The paper includes extensive ablation studies, collaborative gain analysis, role distribution analysis, and scaling experiments that provide valuable insights.
5. **Strong Empirical Results**: Consistent improvements across tasks with statistically significant results on most datasets.

### Weaknesses

1. **Limited Theoretical Foundation**: The paper lacks theoretical analysis of convergence properties, optimality guarantees, or conditions under which the approach should work well. The choice of PSO and the specific algorithmic design appear largely empirical.
2. **Computational Overhead**:
    - The optimization process requires O(n(N+M)) model inferences per iteration, which could be prohibitively expensive for large model pools
    - Limited analysis of computational cost vs. performance trade-offs
    - The speedup strategies (sparsity, dropout) help but may not address fundamental scalability issues
3. **Architecture Constraints**: The requirement that all models share the same architecture for weight optimization significantly limits practical applicability, as real-world scenarios often involve heterogeneous model architectures.
4. **Evaluation Limitations**:
    - Only tested with relatively small models (7B parameters)
    - Limited to 10 expert models - scalability to larger pools unclear
    - Some datasets have small evaluation sets (e.g., GAIA-text with 28 examples)
5. **Methodological Concerns**:
    - The continuous-to-discrete conversion in G-DECODE may lose important structural information
    - PSO hyperparameter sensitivity not thoroughly analyzed
    - The JFK-score formulation could be sensitive to random assignment variations

### Technical Issues

1. **G-DECODE Algorithm**: While novel, the algorithm's reliance on top-p sampling for both end node selection and edge creation introduces stochasticity that may affect reproducibility. The paper would benefit from analysis of this sensitivity.
2. **JFK-Score Robustness**: The individual contribution metric depends on random model assignments. More analysis on the stability and reliability of this metric across different assignment strategies would strengthen the work.

---

> ### Author Rebuttal · Authors · 2025-07-30
>
> We would like to thank the reviewer for their thoughtful comments and feedback.
>
> > The paper lacks theoretical analysis of convergence properties, optimality guarantees, or conditions under which the approach should work well. The choice of PSO and the specific algorithmic design appear largely empirical.
>
> There are theoretical analyses of PSO convergence in evolutionary algorithm research such as [1], while the assumptions made in the classic optimization problems are not applicable to LLMs. For example, the initial particles in classic PSO are often initialized randomly or in a grid, while the initial LMs in the swarm cannot be random/arbitrary (taking a random point in the 7-billion-dimensional space would almost certainly be a failed language model). Instead they are seeded with curated model checkpoints with different training and fine-tuning data mixtures. The utility functions here (often performance on a dataset) are also much more complex than the assumptions made in the analysis of traditional optimization problems.
>
> Given these differences, we decided to take an empirical route to analyze convergence and stability properties. We run H-Swarm for 20 times on three datasets and report its average, worst, best, as well as standard deviation.
>
> |         |  avg  |  max  |  min  |   std  |
> |:-------:|:-----:|:-----:|:-----:|:------:|
> | K-Cross | 0.447 | 0.500 | 0.406 | 0.0279 |
> |   COM2  | 0.587 | 0.617 | 0.560 | 0.0196 |
> | NLGraph | 0.655 | 0.713 | 0.608 | 0.0330 |
>
> Results show that ths standard deviation are all < 0.05, indicating that empirically the algorithm is mostly stable across runs.
>
> > The optimization process requires O(n(N+M)) model inferences per iteration, which could be prohibitively expensive for large model pools. Limited analysis of computational cost vs. performance trade-offs. The speedup strategies (sparsity, dropout) help but may not address fundamental scalability issues
>
> We discuss time/space complexity in Appendix C. Briefly summarize:
>
> Given a pool of $n$ LLMs, we denote the data size behind a utility function as $|f|$ and run H-Swarm with $N$ graphs and $M$ model assignments. The optimization cost is then $O(n(N+M)|f|)$ and the inference cost $O(n)$. As $N$ and $M$ are hyperparameter constants and $|f|$ is a fixed dataset, both costs scale only linearly with the LLM pool size $n$.
>
> As for memory costs, you don’t need to load all $n$ models at once. At the bare minimum, you only need 1 GPU that could load 1 model, do its inference, and switch it out for the next model. At the maximum, you could use $n$ GPUs to load one model per GPU. Any amount of GPU/memory is workable through space-time tradeoff,s and our implementation supports any (#model, #GPU) combinations.
>
> Empirically, it isn’t actually expensive if you refine the implementation. We employ multiprocessing and distributed generation: if multiple GPUs are available, models are separately loaded into memory, and through rotational generation the throughput is very high. If you look at Figure 7: 10 7B models, 1000 data points in f, 5 A40 GPUs, then in 0.43 to 3.02 hours you get a collaboration structure. We personally wouldn’t call 5 GPUs for 3 hours “prohibitively expensive”.
>
> > The requirement that all models share the same architecture for weight optimization significantly limits practical applicability, as real-world scenarios often involve heterogeneous model architectures.
>
> We could run with different model architectures if you move the optimization from the weight space to the logit space. For a pool of n models, each model starts with a one-hot vector v_i = (0, …, 1, …, 0) of size n. The i-th model generates text by decoding from logit distribution $p_i’ = sum_{j=1}^n v_{i,j} p_j$, learning an aggregate distribution of itself and other models. In the beginning, as v_i is one-hot $p_i’ = p_i$ and it only employs itself. As optimization goes, v_i becomes dense and models learn to collaborate by contributing logit distributions.
>
> This type of logit-level collaboration is successful across many scenarios [2,3]. We carried it out for a pool of 5 7b and 5 2b models so the pool contains heterogeneous architectures: Table 7 shows that it works and improves over various other settings of 2b/7b models.
>
> > Only tested with relatively small models (7B parameters); Limited to 10 expert models - scalability to larger pools unclear
>
> Using 7B LMs is common practice for research on this topic. In fact, most of the baseline papers employed 7/8B language models in their experiments when open-access models are employed.
>
> 10 models is a sufficiently large number for research on this topic. For baselines where multiple models are employed: dare used 3, model swarms used 10, ties used 11, greedy soups used 12, pack of llms used 2-6, etc. Optimization/inference cost scales linearly with the amount of models as explained above.
>
> > PSO hyperparameter sensitivity not thoroughly analyzed
>
> We separately change various hyperaprameters into five different candidates, run for 5 times each, and report the standard deviation on two tasks:
>
> |                  | NLGraph | AbstainQA |
> |:----------------:|:-------:|:---------:|
> |      inertia     |  0.012  |   0.023   |
> | cognitive coeff. |  0.010  |   0.015   |
> |   social coeff.  |  0.010  |   0.012   |
> |   repel coeff.   |  0.009  |   0.014   |
> |    step length   |  0.012  |   0.033   |
> |      overall     |  0.011  |   0.018   |
>
> H-Swarm is not sensitive to the major hyperparameters in PSO, with an overall std < 0.02 across runs.
>
> > JFK-Score Robustness: The individual contribution metric depends on random model assignments. The JFK-score formulation could be sensitive to random assignment variations.
>
> The goal *is* for JFK-scores to change when the assignment varies. In this way, by sampling M assignments, we obtain signals about a model’s general helpfulness for a task and how versatile it could be across different roles and contexts.
>
> > G-DECODE Algorithm: While novel, the algorithm's reliance on top-p sampling for both end node selection and edge creation introduces stochasticity that may affect reproducibility. The paper would benefit from analysis of this sensitivity.
>
> To test the impact of algorithmic randomness, we fix the hyperparameters as the best-found set and run H-Swarms for 20 times on three datasets and report its average, worst, best, as well as standard deviation.
>
> |         |  avg  |  max  |  min  |   std  |
> |:-------:|:-----:|:-----:|:-----:|:------:|
> | K-Cross | 0.447 | 0.500 | 0.406 | 0.0279 |
> |   COM2  | 0.587 | 0.617 | 0.560 | 0.0196 |
> | NLGraph | 0.655 | 0.713 | 0.608 | 0.0330 |
>
> Results show that ths standard deviation are all < 0.05, indicating that empirically the algorithm is mostly stable across runs.
>
> For deterministic outcomes, you could employ greedy decoding in G-Decode to always get the same graph structure from any A. Empirically we found that having this randomness actually helps to sample diverse structures and explore more potential collaboration patterns.
>
> [1] Van den Bergh, Frans, and Andries P. Engelbrecht. "A study of particle swarm optimization particle trajectories." Information sciences.
>
> [2] Liu, Alisa, et al. "Tuning Language Models by Proxy."
>
> [3] Mavromatis, Costas, Petros Karypis, and George Karypis. "Pack of LLMs: Model Fusion at Test-Time via Perplexity Optimization."

---

> > ### Comment · Reviewer_cehQ · 2025-08-01
> >
> > Thank you for your comments, which address most of my concerns. I will raise my score accordingly.

---

### Official Review · Reviewer_yVca · 2025-06-23

**Clarity:** 3
**Significance:** 4
**Originality:** 3
**Rating:** 5
**Confidence:** 5

**Summary:**

This paper presents Heterogeneous Swarms (HS), an algorithm that optimizes multi-LLM systems by representing them as DAGs where models exchange messages. Guided by a given utility function, the method uses Particle Swarm Optimization to jointly optimize both the LLM connection structure and their individual parameters. The optimization alternates between a "role-step", which learns the graph topology, and a "weight-step", which adapts model parameters based on a novel "JFK-score" metric designed to quantify each LLM's individual contribution. Through extensive experiments, the proposed method is shown to outperform 17 baselines by an average of 18.5% across 12 diverse tasks, demonstrating substantial collaborative gains.

**Questions:**

1. Could you provide a more detailed comparison of the resource requirements for HS versus key baselines? Specifically, a breakdown of metrics such as total optimization time, cost for optimization and inference (e.g., total tokens processed or GPU hours), would be very helpful for understanding the practical trade-offs involved.
2. The optimization addresses a very high-dimensional search space. How does the framework ensure adequate exploration of the search space within 20 iteration limit to find effective solutions? Was any specific initialization strategy employed beyond random initialization to guide the search?
3. In Table 1, the "Prediction Merge" baseline performs worse than the "Best Single" expert. This is somewhat counterintuitive for an ensemble method. Could the authors offer some insight into why a simple plurality vote might degrade performance?
4. In table 2, the results for "Ours w/o Role" can be interpreted as a dynamic weight optimization method on a fixed graph. This approach still outperforms many dedicated dynamic weight baselines. What is the key algorithmic difference that leads to this superior performance?
5. In appendix, the paper demonstrates that a collaboration of weaker models can outperform a stronger single model. How does the total inference cost of the "HS(2-10)" or "HS(6-10)" swarms compare to the inference cost of the single top-1 model they outperform? Is there a trade-off between collaborative performance gain and increased inference budget?
6. Could you clarify how the utility function $f$ was defined for each of the 12 benchmarks? While accuracy is a likely metric for many, a detailed breakdown in the appendix specifying the evaluation metric for each dataset is needed to enhance the clarity.

**Ethical Concerns:**

["NO or VERY MINOR ethics concerns only"]

**Final Justification:**

I vote for acceptance of the paper. It is a good paper with strong technical contribution and solid empirical validation.

**Limitations:**

Yes. Limitations have been thoroughly discussed.

**Paper Formatting Concerns:**

None.

**Quality:**

4

**Strengths And Weaknesses:**

Strengths:
1. The experiment setup is comprehensive. The authors compare HS against 17 baselines that cover a wide spectrum of approaches, including static/dynamic roles and static/dynamic weights. This rigorous comparison across 12 diverse tasks provides strong evidence for the method's effectiveness.
2. The paper is well-written, with clear explanations of its complex methodology. The analysis is thorough and insightful. For instance, the ablation study disentangles the relative importance of roles and weights for different task types, justifying the joint optimization approach.  Furthermore, the analysis of "collaborative gains" provides a quantitative justification for the use of multi-LLM systems by showing that the swarm can solve problems that no single constituent model can.
3. While the concept of multi-LLM collaboration is not new, the explicit formulation of this as a joint optimization problem over graph structures and model parameters is a significant contribution. The methods and metrics introduced, such as the iterative two-step optimization process and the JFK-score for credit assignment, are well-grounded in the problem they aim to solve. This provides a flexible yet powerful framework for designing multi-LLM systems.

Weaknesses:
1. The primary limitation of HS is its computational expense. The optimization process involves numerous inferences across a swarm of graphs and models. While the authors acknowledge this and propose sparsity and dropout strategies to mitigate the cost, the optimization phase still remains expensive.
2. The weight-step of the algorithm, in its default implementation, requires all LLMs in the pool to share the same model architecture to ensure they share the same parameter space for optimization. While the authors suggest potential workarounds, this is a notable constraint that limits the diversity of truly "heterogeneous" models that can be incorporated, especially proprietary models which cannot be fine-tuned.
3. The role-step optimizes continuous adjacency matrices, which is a very high-dimensional search space ($n^2$ for $n$ models). While PSO is suited for non-differentiable problems, its effectiveness can degrade in such large search spaces. The paper could benefit from a deeper discussion on why PSO is the right choice over other black-box optimization techniques (e.g., evolutionary strategies, Bayesian optimization) and how it scales with an increasing number of LLMs.

---

> ### Author Rebuttal · Authors · 2025-07-30
>
> We would like to thank the reviewer for their thoughtful comments and feedback.
>
> > The primary limitation of HS is its computational expense. The optimization process involves numerous inferences across a swarm of graphs and models. While the authors acknowledge this and propose sparsity and dropout strategies to mitigate the cost, the optimization phase still remains expensive.
>
> We discuss time/space complexity in Appendix C. Briefly summarize:
>
> Given a pool of $n$ LLMs, we denote the data size behind a utility function as $|f|$ and run H-Swarm with $N$ graphs and $M$ model assignments. The optimization cost is then $O(n(N+M)|f|)$ and the inference cost $O(n)$. As $N$ and $M$ are hyperparameter constants and $|f|$ is a fixed dataset, both costs scale only linearly with the LLM pool size $n$.
>
> As for memory costs, you don’t need to load all $n$ models at once. At the bare minimum, you only need 1 GPU that could load 1 model, do its inference, and switch it out for the next model. At the maximum, you could use $n$ GPUs to load one model per GPU. Any amount of GPU/memory is workable through space-time tradeoff,s and our implementation supports any (#model, #GPU) combinations.
>
> Empirically, it isn’t actually expensive if you refine the implementation. We employ multiprocessing and distributed generation: if multiple GPUs are available, models are separately loaded into memory, and through rotational generation the throughput is very high. If you look at Figure 7: 10 7B models, 1000 data points in f, 5 A40 GPUs, then in 0.43 to 3.02 hours you get a collaboration structure. We personally wouldn’t say 5 GPUs for 3 hours is too “expensive”.
>
> > The weight-step of the algorithm, in its default implementation, requires all LLMs in the pool to share the same model architecture to ensure they share the same parameter space for optimization. While the authors suggest potential workarounds, this is a notable constraint that limits the diversity of truly "heterogeneous" models that can be incorporated, especially proprietary models which cannot be fine-tuned.
>
> *Open models with different architectures*
>
> We move the optimization from the weight space to the logit space. For a pool of n models, each model starts with a one-hot vector v_i = (0, …, 1, …, 0) of size n. The i-th model generates text by decoding from logit distribution $p_i’ = sum_{j=1}^n v_{i,j} p_j$, learning an aggregate distribution of itself and other models. In the beginning, as v_i is one-hot $p_i’ = p_i$ and it only employs itself. As optimization goes, v_i becomes dense and models learn to collaborate by contributing logit distributions.
>
> This type of logit-level collaboration is successful across many scenarios [1,2]. We carried it out for a pool of 5 7b and 5 2b models so the pool contains heterogeneous architectures: Table 7 shows that it works and improves over various other settings of 2b/7b models.
>
> *Closed models that cannot be finetuned*
>
> H-Swarm is compatible with closed models too: run inference on them as usual, in the weight-step just skip the PSO weight updates for them and update other open models only.
>
> > The role-step optimizes continuous adjacency matrices, which is a very high-dimensional search space (for models). While PSO is suited for non-differentiable problems, its effectiveness can degrade in such large search spaces. The paper could benefit from a deeper discussion on why PSO is the right choice over other black-box optimization techniques (e.g., evolutionary strategies, Bayesian optimization) and how it scales with an increasing number of LLMs. The optimization addresses a very high-dimensional search space. How does the framework ensure adequate exploration of the search space within 20 iteration limit to find effective solutions? Was any specific initialization strategy employed beyond random initialization to guide the search?
>
> The search space is high-dimensional, but swarm for LLMs doesn’t need to explore the vast search space compared to traditional optimization problems:
>
> |    Setting    |                               Initialization                               |                            Exploration                            |
> |:-----------:|:--------------------------------------------------------------------------:|:-----------------------------------------------------------------:|
> | traditional |                           many and mostly random                           |   extensive and it is unclear where in the search space is good   |
> |  LLM world  | a few, but they are already of good quality since they are LLM checkpoints | mostly the convex hull and linear mesh of the initial checkpoints |
>
> We use t-sne for dimensionality reduction to plot the movement of LLMs in the 20 iterations (couldn't show figures here). We find that they do not (and don’t need to) go all over the place, instead they converge mostly within the convex hull of models that are already of moderate quality, just not fully adapted to this task.
>
> > Could you provide a more detailed comparison of the resource requirements for HS versus key baselines? Specifically, a breakdown of metrics such as total optimization time, cost for optimization and inference (e.g., total tokens processed or GPU hours), would be very helpful for understanding the practical trade-offs involved.
>
> We summarize the optimization/inference complexity in Table 4 in the appendix, to recap:
>
> |      Method      | Optimization | Inference |
> |:----------------:|:------------:|:---------:|
> |    best single   |       /      |    O(1)   |
> | prediction merge |       /      |    O(n)   |
> |   static weight  |       /      |    O(1)   |
> |  dynamic weight  |     O(n)     |    O(1)   |
> |    static role   |       /      |    O(n)   |
> |   dynamic role   |     O(nN)    |    O(n)   |
> |       ours       |   O(n(N+M))  |    O(n)   |
>
> Since our approach does more (role AND weight) naturally the optimization cost is higher, while the inference time is on par with the stronger baselines.
>
> For an empirical cost, we present the time (in hours) using A40 GPUs for a few approaches:
>
> |      Method      | Optimization (per iter.) | Inference |
> |:----------------:|:------------------------:|:---------:|
> |    Best Single   |             /            |   0.012   |
> | Prediction Merge |             /            |   0.138   |
> |    Greedy Soup   |           1.68           |   0.012   |
> |   Model Swarms   |           1.83           |   0.014   |
> |    Meta-Agent    |           2.07           |   0.264   |
> |       GNNs       |           1.73           |   0.185   |
> |       ours       |           3.02           |   0.204   |
>
> There are many complexities behind this empirical time: some approaches only need 1 GPU and approaches like ours could scale well with multiple GPUs using multiprocessing; approaches like Meta-Agent require black-box models to guide structure designs and introduce additional latency, etc. Overall, our approach has a reasonable cost, could be further reduced with strategies in Table 3, and leads to better utility through the co-evolution of role and weight.
>
> > In Table 1, the "Prediction Merge" baseline performs worse than the "Best Single" expert. This is somewhat counterintuitive for an ensemble method. Could the authors offer some insight into why a simple plurality vote might degrade performance?
>
> This indicates that “truth is in the hands of the few”: because the models are diverse (exposure to different domains, tasks, etc.), without any adaptation only a few (or even just one in some cases) could solve a task-specific problem. The plurality vote of the many is then often incorrect compared to the model that best specializes in this task.
>
> This highlights the importance of the weight-step: to prepare models for the current task in question and adapt their existing skill set to what would be helpful for this task and collaboration.
>
> > In table 2, the results for "Ours w/o Role" can be interpreted as a dynamic weight optimization method on a fixed graph. This approach still outperforms many dedicated dynamic weight baselines. What is the key algorithmic difference that leads to this superior performance?
>
> This is because swarm intelligence and PSO is a strong solution for weight optimization. In Table 1, model swarms is very often the best weight-based approach. While “ours w/o role” uses different utility signals for weight updates (JFK-score, as opposed to individual model performance), they both operate with PSO. By additionally having the role orchestration step, we further outperform Model Swarms.
>
> > In appendix, the paper demonstrates that a collaboration of weaker models can outperform a stronger single model. How does the total inference cost of the "HS(2-10)" or "HS(6-10)" swarms compare to the inference cost of the single top-1 model they outperform? Is there a trade-off between collaborative performance gain and increased inference budget?
>
> If the inference cost of best single is x, then HS(2-10) is ~9x and HS(6-10) ~5x. Yes there is a trade-off: weaker models, if compute allows to be used in collaboration, could become strong multi-model systems.
>
> > Could you clarify how the utility function was defined for each of the 12 benchmarks? While accuracy is a likely metric for many, a detailed breakdown in the appendix specifying the evaluation metric for each dataset is needed to enhance the clarity.
>
> We provide details in Appendix D:
>
> MMLU-pro, Knowledge Crosswords, COM2, Normad, AgentBench-KG, AbstainQA, and WoW are evaluated in multiple-choice settings. GSM8k, NLGraph, and GAIA-text are evaluated via exact match. AgentBench-LTP and Qasper are evaluated by Gemini-1.5-pro for gold answer similarity on a scale of 1 to 10, normalized to 0 to 1.
>
> [1] Liu, Alisa, et al. "Tuning Language Models by Proxy."
>
> [2] Mavromatis, Costas, Petros Karypis, and George Karypis. "Pack of LLMs: Model Fusion at Test-Time via Perplexity Optimization."

---

> > ### Comment · Reviewer_yVca · 2025-08-01
> > **Thanks for the Rebuttal**
> >
> > Thank you for the detailed and thoughtful rebuttal. I appreciate the strong technical contribution and solid empirical validation presented in the paper. The rebuttal provides valuable clarification. I find the discussion on the convex hull and the constrained movement of models insightful. I encourage the authors to include the t-SNE visualizations and related analyses in the appendix of the final version, as they would greatly enhance understanding of the optimization dynamics. Overall, I believe this is a good paper and strongly vote for acceptance.

---

### Official Review · Reviewer_wwh1 · 2025-06-24

**Clarity:** 3
**Significance:** 2
**Originality:** 2
**Rating:** 4
**Confidence:** 3

**Summary:**

This paper aims to use multiple LLMs to solve a task by organizing them as the nodes of a DAG and calling them in the topological order where the output of the parents of a node becomes part of that node’s input.
The paper employs the technique of Particle Swarm Optimization (PSO) to optimize the DAG topology and the LLMs’ weights for the given task.

**Questions:**

In addition to the questions in the Strengths And Weaknesses section:

Q1. (Line 103) What's x_best? Is it the same as g (the global best)?

Q2. Isn't that the DAG itself already has an assignment? If so, why do we “randomly select an LLM for each position” (Line 145)?

Q3. How difficult is it to combine a prior dynamic weight technique vs a prior dynamic role?

Q4. Are the optimization rounds (role-step and weight-step) performed independently for every dataset or datapoint in Table 1?

Q5. Why can the proposed method alleviate prompt engineering issues? What prompts does the proposed method use? How about the baselines

**Ethical Concerns:**

["NO or VERY MINOR ethics concerns only"]

**Final Justification:**

Please see my comment.

**Limitations:**

Limitations are discussed in the appendix.

**Paper Formatting Concerns:**

None.

**Quality:**

2

**Strengths And Weaknesses:**

**Strengths**
The paper addresses a well-motivated problem that is worth more attention.
It is overall written with good clarity and easy to follow.
The empirical improvement seems significant.
The paper provides analysis on the learned roles.

**Weaknesses**
W1. The G-decode algorithm is not well justified. In what sense does it use matrix $A$ as edge probabilities? What is top-p sampling and why is it used here? And there are DAG parameterization techniques that automatically ensure acyclicility:
Charpentier, Bertrand, Simon Kibler, and Stephan Günnemann. "Differentiable DAG Sampling." ICLR 2022.

W2. JFK-score is inherently limiting because it averages out an individual LLM’s performance across many teams. In general, an LLM as a team member performs well only with certain teammates in a certain team. Ideally we would like to identify the best team as a whole. Such limitations are not discussed in the paper.

W3. The innovation seems incremental from Model Swarms [26], which already used PSO for weight update. The paper does not clearly describe the difference from [26].

W4. Using PSO for weight update is confusing.
PSO tries to identify a single $x$, but ideally the multiple LLMs should have their own best weight. It is unmotivated to use PSO for the weight update.
Moreover, PSO is only applicable when the LLMs have the same architecture, but in general an multi-LLM system involves LLMs of various kinds.

W5. It is unclear how meaningful it is to compare with the chosen baselines.
How many baselines do multi-round optimization as the proposed method does?
If prior works have done either dynamic weight or dynamic role, can we do a baseline that does both by combining two prior works (e.g., Model Swarms + GPT-Swarm), which could serve as a stronger baseline to test the effectiveness of the proposed role-step/weight-step?

---

> ### Author Rebuttal · Authors · 2025-07-30
>
> We would like to thank the reviewer for their thoughtful comments and feedback.
>
> > The G-decode algorithm is not well justified. In what sense does it use matrix as edge probabilities? What is top-p sampling and why is it used here? And there are DAG parameterization techniques that automatically ensure acyclicility.
>
> For a pool of n models, adjacency matrix A is size $n \times n$. $a_{ij}$ denotes the likelihood of a directed edge from model i to j. (lines 121-122)
>
> Top-p sampling is a widely employed strategy for decoding, particularly in language models [1]. Instead of greedily selecting the $argmax$ element or sampling from the whole distribution, top-p sampling only considered the highest-prob candidates with a cumulative probability of p. This strikes a nice balance between randomness and quality, in our specific case it helps to decode multiple good-quality structures from A to better explore diverse collaboration patterns.
>
> Thank you for the important reference. We are citing it and adding “other strategies such as Charpentier et al. could also produce DAGs from A” in the revised paper.
>
> > JFK-score is inherently limiting because it averages out an individual LLM’s performance across many teams. In general, an LLM as a team member performs well only with certain teammates in a certain team. Ideally we would like to identify the best team as a whole. Such limitations are not discussed in the paper.
>
> Using the team analogy: the weight-step improves the individual while the role-step improves the team. By using JFK-score in the weight-step, models become versatile for any potential role in the given task. These models then form the best team in the role-step through graph optimization and orchestration. JFK-score fulfills this motivation by providing a signal of whether the model is generally helpful for a given task, not role-dependent or placement-dependent.
>
> > The innovation seems incremental from Model Swarms [26], which already used PSO for weight update. The paper does not clearly describe the difference from [26]. Using PSO for weight update is confusing. PSO tries to identify a single but ideally the multiple LLMs should have their own best weight. It is unmotivated to use PSO for the weight update. Moreover, PSO is only applicable when the LLMs have the same architecture, but in general an multi-LLM system involves LLMs of various kinds.
>
> We summarize the key differences:
>
> |  Method    |                                       Optimization                                       |                 Inference                 |                                             Applications                                             |
> |:----:|:----------------------------------------------------------------------------------------:|:-----------------------------------------:|:----------------------------------------------------------------------------------------------------:|
> | [26] |                             weights of individual models only                            |               a single model              |                                tasks where a single model could tackle                               |
> | ours | different utility f for weight & new role-step that orchestrates model collaboration | a DAG of multiple models in collaboration | above, plus agentic and especially multi-agent scenarios where the collaboration is helpful/required |
>
> Different models actually have different $x$ because their movement is guided by different velocity $v$: if you look at the velocity update equation between line 95 and 96: signals $g$ and $g_w$ are shared across the swarm while $v_i$ and $p_i$ are model-specific (as evident by the subscript i). This allows multiple models to separately explore while having something in common, i.e. the task that models are collaborating for.
>
> *PSO could run with different model architectures* if you move the optimization from the weight space to the logit space. For a pool of n models, each model starts with a one-hot vector v_i = (0, …, 1, …, 0) of size n. The i-th model generates text by decoding from logit distribution $p_i’ = sum_{j=1}^n v_{i,j} p_j$, learning an aggregate distribution of itself and other models. In the beginning, as v_i is one-hot $p_i’ = p_i$ and it only employs itself. As optimization goes, v_i becomes dense and models learn to collaborate by contributing logit distributions.
>
> This type of logit-level collaboration is successful across many scenarios [2,3, and also [26] itself]. We carried it out for a pool of 5 7b and 5 2b models so the pool contains heterogeneous architectures: Table 7 shows that it works and improves over various other settings of 2b/7b models.
>
> > It is unclear how meaningful it is to compare with the chosen baselines. How many baselines do multi-round optimization as the proposed method does? If prior works have done either dynamic weight or dynamic role, can we do a baseline that does both by combining two prior works (e.g., Model Swarms + GPT-Swarm), which could serve as a stronger baseline to test the effectiveness of the proposed role-step/weight-step? How difficult is it to combine a prior dynamic weight technique vs a prior dynamic role?
>
> Multi-round: dynamic weight and dynamic role baselines, 10 in total, are all multi-round, which repeatedly evaluate on f. We use 20 max iterations across all approaches, including ours.
>
> Both role and weight: we are the first to the best of our knowledge, as this is one of our contributions.
>
> There won’t be a natural way to combine a role-only and weight-only approach as they have different utility and goals. We reconcile this gap through G-Decode, through JFK-score, through the swarm methodology, to present a seamless role-AND-weight approach where mutual information between the two steps reinforces each other. But if you have to, you could force the same goal upon these two approaches. We try this out where both approaches are strictly optimized no matter single-model or multi-model:
>
> |                | K-Cross | AB-ltp | AbstainQA |
> |:--------------:|:-------:|:------:|:---------:|
> |  Model Swarms  |  0.428  |  0.135 |   0.175   |
> |    GPT-Swarm   |  0.320  |  0.134 |   0.023   |
> | forced combine |  0.376  |  0.109 |   0.112   |
> |      ours      |  0.450  |  0.215 |   0.220   |
>
> It isn’t better than ours, in fact it is even worse than the two component methods on AB-ltp, potentially due to the multi-agent nature of this task. We believe our role-AND-weight approach is uniquely effective beyond forcibly gluing two parts together.
>
> > (Line 103) What's x_best? Is it the same as g (the global best)?
>
> x_best is the best-found location of this step, while global best g is the best-found across all steps. We will add this sentence to line 104.
>
> > Isn't that the DAG itself already has an assignment? If so, why do we “randomly select an LLM for each position” (Line 145)?
>
> Yes, the default assignment to a DAG is “model i as node i” and each model appears exactly once. We do new assignments to 1) better explore different collaboration possibilities and 2) allow one model to potentially appear multiple times in the DAG while models unhelpful to the task don’t need to appear at all.
>
> > Are the optimization rounds (role-step and weight-step) performed independently for every dataset or datapoint in Table 1?
>
> Yes, it is per dataset for a task adaptation setting. We will state this in Section 4, line 179.
>
> > Why can the proposed method alleviate prompt engineering issues? What prompts does the proposed method use? How about the baselines
>
> In many multi-agent/model approaches, you have to manually design prompts for models to play different roles. For example, in one baseline Meta-Agent, one of the many prompts (and there are many more like this) is as follows and I’m sure has gone through many manual iterations:
>
> ```
> You are an expert machine learning researcher testing various agentic systems. Your objective is to
> design building blocks such as prompts and workflows within these systems to solve complex tasks.
> Your aim is to design an optimal agent performing well on [Brief Description of the Domain].
> [Framework Code]
> [Output Instructions and Examples]
> [Discovered Agent Archive] (initialized with baselines, updated at every iteration)
> # Your task
> You are deeply familiar with prompting techniques and the agent works from the literature. Your goal is
> to maximize the specified performance metrics by proposing interestingly new agents.
> Observe the discovered agents carefully and think about what insights, lessons, or stepping stones can
> be learned from them.
> Be creative when thinking about the next interesting agent to try. You are encouraged to draw inspiration
> from related agent papers or academic papers from other research areas.
> Use the knowledge from the archive and inspiration from academic literature to propose the next interesting agentic system design.
> THINK OUTSIDE THE BOX.
> ```
>
> In our case (Table 9), the prompt is only:
>
> ```
> “Please answer the following question with the help of previous responses, feel free to ignore wrong or unhelpful Responses.”
> ```
>
> Instead of manual design, we count on optimization so models can automatically learn and discover what I should do/what my role is, given the responses of previous models. Figure 4 illustrates that models do discover roles such as divide and conquer, feedback, refine, etc., despite not being specifically asked for in the prompt.
>
> [1] Holtzman, Ari, et al. "The Curious Case of Neural Text Degeneration."
>
> [2] Liu, Alisa, et al. "Tuning Language Models by Proxy."
>
> [3] Mavromatis, Costas, Petros Karypis, and George Karypis. "Pack of LLMs: Model Fusion at Test-Time via Perplexity Optimization."

---

> ### Comment · Reviewer_wwh1 · 2025-08-02
>
> Thank you for your response. I am raising my score to 4.
>
> Some of my concerns are still not addressed.
> - My question in W1 was: For the graph sampled by G-decode, does edge $(i, j)$ exist with probability $a_{ij}$?
> - For W1 again, is top-p necessary to get good performance? Did authors try standard sampling?
> - I think my comment on JFK-score was correct. It was just repeating lines 150-151. I will keep my opinion on this point.
> - Difference of this paper vs [26] on weight update is still unclear to me. Do both use PSO-based weight update? If yes, what's the difference in the update rule?
> - The practice of “randomly select an LLM for each position” (Line 145) is still confusing. The entire algoirthm is designed for “model i as node i”, but the assignment is not. This mismatch seems major.
>
> I am not familiar with the literature related to the experiments, but the proposed method does seem to yield significant improvement. Yet the concerns above justify my score.
>
> Some of my concerns are largely addressed:
> - Your answer helps me understand the motivation of using PSO for weight update. Thanks.
> - The logit technique was not described in the original submission. It would be helpful to include this.
> - The prompt used is simple.

---

> > ### Author Response · Authors · 2025-08-02
> >
> > Thank you for raising the score! Following up on these points:
> >
> > > My question in W1 was: For the graph sampled by G-decode, does edge $(i,j)$ exist with probability $a_{jk}$?
> >
> > No. In this way, acyclic wouldn't be guaranteed. These probabilities exist so that when a new node $a_i$ (and edge $a_i$ -> ?) is considered, one node could be sampled (from the existing nodes in the graph) based on the likelihood $a_{i,j}$.
> >
> > > For W1 again, is top-p necessary to get good performance? Did authors try standard sampling?
> >
> > Yes. In standard sampling, it is possible to sample low-likelihood edges which would not be a good collaboration between two models. This could have cascading effects: if one edge between two models is not desirable, it could lead to failures in future steps. Having top-p sampling prevents low-prob edges from being sampled, preventing this. We find empirical results similar to patterns in [1], that a low-prob "bad" edge might derail the system.
> >
> > > Difference of this paper vs [26] on weight update is still unclear to me. Do both use PSO-based weight update? If yes, what's the difference in the update rule?
> >
> > They both use PSO for weight update, but the utility function is different. [26]'s f is "an individual model's performance on a task" while in ours f is JFK-score measuring "overall contribution to a task across multiple collaboration systems".
> >
> > > The practice of “randomly select an LLM for each position” (Line 145) is still confusing. The entire algoirthm is designed for “model i as node i”, but the assignment is not. This mismatch seems major.
> >
> > The role-step assumes “model i as node i” to produce a graph structure with one model assigned once, but in the weight-step it is not: different assignments are intentionally created to explore how might models perform in different places of the graph structure, allowing for helpful models to appear more than once and unhelpful models to not appear at all. If we only have "model i as node i" without trying out assignments, models unrelated/unadapted for the given task would also participate and might harm the system.
> >
> > [1] Holtzman, Ari, et al. "The Curious Case of Neural Text Degeneration."

---

### Official Review · Reviewer_zL7f · 2025-07-02

**Clarity:** 3
**Significance:** 3
**Originality:** 3
**Rating:** 4
**Confidence:** 2

**Summary:**

This paper focuses on multi-LLM systems and figures out the fixed-weight/fixed-role limitations in the existing methods. To overcome these limitations, the authors propose an algorithm named HETEROGENEOUS SWARMS to jointly optimizes the collaborative structure (a DAG) and individual model weights via Particle Swarm Optimization (PSO). The proposed method shows impressive outperformance over numerous baselines across 12 diverse tasks, with analysis confirming substantial collaborative gains. Though I concern about its generalizability and computation cost, this work presents a powerful framework for designing adaptive multi-LLM systems and marks a valuable contribution to the field.

**Questions:**

1. It is a little counter-intuitive for me that BEST SINGLE usually achieves better performance than PRED. MERGE as shown in Table 1. Can you briefly explain about it?
2. Is the utility function optimized per task or not? This point is ambiguous in the paper.

**Ethical Concerns:**

["NO or VERY MINOR ethics concerns only"]

**Final Justification:**

The author's rebuttal clarifies some points and makes me understand this paper better. I believe my current score (4) has shown my positive feedback on your paper and will keep it.

**Limitations:**

yes

**Paper Formatting Concerns:**

N.A.

**Quality:**

3

**Strengths And Weaknesses:**

* Strength
1. The paper formulate multi-LLM system problem-solving as jointly optimizing model roles (DAG) and weights. This novel modeling provides a structured alternative to prior methods which often treated these components separately or relied on manual heuristics, thereby creating a more comprehensive optimization target.
2. The proposed algorithm offers a practical solution to a complex search problem by employing PSO combined with two specific components—G-decode for graph generation and the JFK-score for credit assignment. These methods successfully navigate the high-dimensional, non-differentiable search space of possible graph structures and model parameters.
3. The authors conduct a broad set of experiments across 12 different tasks and compare their method against 17 baselines. The consistent performance improvements demonstrated the effectiveness of this algorithm. Furthermore, the analysis and ablation study is insightful and confirms the necessity of each design modules.

* Weakness

IMO the topological structures of the multiple LLMs are task-dependent, i.e., differ task by task.  While the appendix presents a single experiment on generalization (Table 6), this evidence is insufficient to fully support the claim of adaptability. The experiment shows generalization from one knowledge-based task (K-Cross) to another similar one (WikiDYK). This demonstrates in-domain generalization but does not address the more challenging case of out-of-domain generalization (e.g., a system optimized for reasoning being applied to an agentic task). The high degree of task-specific optimization inherent in the method raises concerns about how brittle these specialized systems might be when faced with novel problem types.

---

> ### Author Rebuttal · Authors · 2025-07-30
>
> We would like to thank the reviewer for their thoughtful comments and feedback.
>
> > IMO the topological structures of the multiple LLMs are task-dependent, i.e., differ task by task. While the appendix presents a single experiment on generalization (Table 6), this evidence is insufficient to fully support the claim of adaptability. The experiment shows generalization from one knowledge-based task (K-Cross) to another similar one (WikiDYK). This demonstrates in-domain generalization but does not address the more challenging case of out-of-domain generalization (e.g., a system optimized for reasoning being applied to an agentic task). The high degree of task-specific optimization inherent in the method raises concerns about how brittle these specialized systems might be when faced with novel problem types.
>
> Yes, it is task-dependent: we believe there won’t be general-purpose model/agent collaboration structures that fit every task and context, so what we did is to provide an optimization algorithm to discover what works for each task. This is in line with general practice (*general* method to discover *task-specific* collaboration patterns). [1-2] We don’t think demanding a task-specific structure to work when out-of-domain is fair, as there won’t be a single one-size-fits-all structure.
>
> What might be possible is grouping tasks by domains (e.g. knowledge, reasoning, etc.) and hope to discover structures that work across this domain: we demonstrated that in the knowledge domain in Table 6.
>
> > It is a little counter-intuitive for me that BEST SINGLE usually achieves better performance than PRED. MERGE as shown in Table 1. Can you briefly explain about it?
>
> This indicates that “truth is in the hands of the few”: because the models are diverse (exposure to different domains, tasks, etc.), without any adaptation only a few (or even just one in some cases) could solve a task-specific problem. The plurality vote of the many is then often incorrect compared to the model that best specializes in this task.
>
> This highlights the importance of the weight-step: to prepare models for the current task in question and adapt their model weights and skill set to what would be helpful for this task and collaboration.
>
> > Is the utility function optimized per task or not? This point is ambiguous in the paper.
>
> Yes, it is per task for a task adaptation setting. We will state this in Section 4, line 179.
>
> [1] Zhuge, Mingchen, et al. "Gptswarm: Language agents as optimizable graphs." ICML.
>
> [2] LangGraph. https://www.langchain.com/langgraph

---

> > ### Comment · Reviewer_zL7f · 2025-08-09
> >
> > Thanks for your reply. It addresses my concern.
> >
> > I believe my current score (4) has shown my positive feedback on your paper and will keep it.

---

### Official Review · Reviewer_y4wm · 2025-07-03

**Clarity:** 3
**Significance:** 3
**Originality:** 3
**Rating:** 4
**Confidence:** 3

**Summary:**

This paper introduces Heterogeneous Swarms, an algorithm that automatically designs multi-LLM systems by jointly optimizing two key dimensions: (i) model roles, represented as a directed acyclic graph (DAG) defining inter-model communication, and (ii) model weights, quantifying each LLM’s contribution. The method alternates between a role-step (discovering the optimal DAG via particle swarm optimization over continuous adjacency matrices) and a weight-step (estimating individual model contributions via a proposed JFK-score and optimizing weights accordingly). The approach is tested on 12 diverse tasks, outperforming 17 strong baselines with an 18.5% average improvement, and demonstrating consistent collaborative gains from heterogeneous multi-LLM systems.

**Questions:**

1. Could the authors clarify how G-DECODE ensures DAG validity in extreme cases where adjacency probabilities are nearly uniform? Does this lead to highly variable graph structures between runs?

2. How does the method perform if the initial pool of LLMs is not diverse (e.g., identical models)? Could the authors quantify the dependency on model diversity in more detail?

3. What happens when the utility function f is noisy or discontinuous? Does PSO still converge reliably?

4. Could the authors discuss scalability: what are the computational and memory costs when the LLM pool size grows beyond 10 or 20 experts?

5. Are there practical strategies (e.g., early stopping heuristics) to reduce the potentially high inference cost during optimization without losing significant performance?

**Ethical Concerns:**

["NO or VERY MINOR ethics concerns only"]

**Final Justification:**

address most of my concerns

**Limitations:**

yes

**Paper Formatting Concerns:**

No major formatting issues observed; the paper appears to comply with NeurIPS 2025 formatting guidelines.

**Quality:**

3

**Strengths And Weaknesses:**

*Strengths*

1. The paper is well-written with clear explanations of complex components (e.g., G-DECODE, JFK-score). Figures (e.g., Figure 2) are illustrative and help convey the method’s workflow.

2. The framing of roles as learnable DAG structures optimized with swarm intelligence is novel.  Combining both roles and weights in a unified optimization loop is a fresh contribution compared to prior fixed-role or fixed-weight strategies.

3. The paper advances the field of multi-LLM collaboration by addressing both flexible role discovery and weight adaptation—two aspects often studied separately. The empirical gains suggest substantial practical impact.

*Weaknesses*

1.  The method’s reliance on repeated LLM inference during optimization could make it impractical for very large models without distributed compute.

2. The paper builds on particle swarm optimization in a relatively straightforward way; while well-applied, PSO itself is not a new method.

3. A few sections, especially the PSO update rule and G-DECODE sampling process, could benefit from more intuitive examples or step-by-step illustrations.

---

> ### Author Rebuttal · Authors · 2025-07-30
>
> We would like to thank the reviewer for their thoughtful comments and feedback.
>
> > The method’s reliance on repeated LLM inference during optimization could make it impractical for very large models without distributed compute. Could the authors discuss scalability: what are the computational and memory costs when the LLM pool size grows beyond 10 or 20 experts? Are there practical strategies (e.g., early stopping heuristics) to reduce the potentially high inference cost during optimization without losing significant performance?
>
> We discuss time/space complexity in Appendix C. Briefly summarize:
>
> Given a pool of $n$ LLMs, we denote the data size behind a utility function as $|f|$ and run H-Swarm with $N$ graphs and $M$ model assignments. The optimization cost is then $O(n(N+M)|f|)$ and the inference cost $O(n)$. As $N$ and $M$ are hyperparameter constants and $|f|$ is a fixed dataset, both costs scale linearly with the LLM pool size $n$.
>
> As for memory costs, you don’t need to load all $n$ models at once. At the bare minimum, you only need 1 GPU that could load 1 model, do its inference, and switch it out for the next model. At the maximum, you could use $n$ GPUs to load one model per GPU. Any amount of GPU/memory is workable through space-time tradeoff, and our implementation supports any (#model, #GPU) combinations.
>
> We implement a quick early-stopping heuristic: for the k-th model’s response, we parse it to determine whether an answer is reached (“the answer is / the final answer is / the correct answer / the solution is / …”) and exit if found, stopping future models from further critiquing and refining the answer. This lowers inference cost from O(n) to O(k): we report performance and the average k across three tasks:
>
> |                | K-Cross | NLGraph | AbstainQA |
> |:--------------:|:-------:|:-------:|:---------:|
> | full inference |  0.450  |  0.660  |   0.220   |
> | early stopping |  0.437  |  0.591  |   0.205   |
> |     avg. k     |   6.38  |   4.53  |    7.22   |
>
> We observe minor drops on K-Cross and AbstainQA with a 32% speedup, while on the reasoning task NLGraph performance dropped more with a higher 55% speedup. This suggests that different tasks might require different early stopping strategies, which we will further explore in the future.
>
> > Could the authors clarify how G-DECODE ensures DAG validity in extreme cases where adjacency probabilities are nearly uniform? Does this lead to highly variable graph structures between runs?
>
> No matter what the adjacency probs are, we use top-p sampling to sample one node at a time and produce an (sampled_node -> existing_node_in_the_graph) edge, so it will always be acyclic regardless of underlying probabilities of selecting the node. If A is near uniform, there could be great randomness in graph decoding: to curb this you could use p=0 (greedy decoding) so that the produced graphs are deterministic. In practice, we find that optimized A is almost never near uniform and enabling this variation and randomness in graphs actually helps to sample and explore diverse model collaboration patterns.
>
> > How does the method perform if the initial pool of LLMs is not diverse (e.g., identical models)? Could the authors quantify the dependency on model diversity in more detail?
>
> Figure 5, leftmost, presents the setting where all 10 models are identical (i.e. the same model checkpoint, or “best single” in Table 1). We observe steady upward trends from this no-diversity setting to the most-diversity setting where the 10 models are all different: on K-Cross performance improved from 27.1 to 45.0 and on AbstainQA from 9.4 to 22.0, indicating major improvements thanks to the collaboration of diverse model checkpoints.
>
> > What happens when the utility function f is noisy or discontinuous? Does PSO still converge reliably?
>
> We simulate noise in $f$ by randomly flipping 5% and 10% labels in the development set (used in $f$ and optimization) while keeping the test set intact.
>
> |         | 0% noise | 5% noise | 10% noise |
> |:-------:|:--------:|:--------:|:---------:|
> | K-Cross |   0.450  |   0.418  |   0.421   |
> |  Normad |   0.588  |   0.559  |   0.544   |
>
> There is a slight drop from 0% to 5% and a negligible change from 5% to 10%, indicating that the PSO methodology is moderately robust to dataset noise.

---

> > ### Comment · Reviewer_y4wm · 2025-08-06
> >
> > Thank you for your reply. I will maintain my positive rating.

---

### Decision · Program_Chairs · 2025-09-17

**Decision:**

Accept (poster)

**Comment:**

This paper proposes Heterogeneous Swarms, an algorithm for optimizing multi-LLM systems by representing them as directed acyclic graphs (DAGs). The proposed algorithm contains two iterative steps: (1) a role step, which learns the DAGs for inputs/outputs flows, and (2) a weight step, which estimates individual contributions and optimizes weights accordingly. Experimental results demonstrate the effectiveness of the proposed approach. After the rebuttal, all reviewers leaned toward acceptance.